# TAAC: Temporally Abstract Actor-Critic for Continuous Control

**Haonan Yu, Wei Xu, Haichao Zhang**
Horizon Robotics
Cupertino, CA 95014
{haonan.yu,wei.xu,haichao.zhang}@horizon.ai

## Abstract

We present temporally abstract actor-critic (TAAC), a simple but effective off-policy RL algorithm that incorporates closed-loop temporal abstraction into the actor-critic framework. TAAC adds a second-stage binary policy to choose between the previous action and a new action output by an actor. Crucially, its "act-or-repeat" decision hinges on the actually sampled action instead of the expected behavior of the actor. This post-acting switching scheme let the overall policy make more informed decisions. TAAC has two important features: a) persistent exploration, and b) a new compare-through Q operator for multi-step TD backup, specially tailored to the action repetition scenario. We demonstrate TAAC's advantages over several strong baselines across 14 continuous control tasks. Our surprising finding reveals that while achieving top performance, TAAC is able to "mine" a significant number of repeated actions with the trained policy even on continuous tasks whose problem structures on the surface seem to repel action repetition. This suggests that aside from encouraging persistent exploration, action repetition can find its place in a good policy behavior. Code is available at https://github.com/hnyu/taac.

## 1 Introduction

Deep reinforcement learning (RL) has achieved great success in various continuous action domains such as locomotion and manipulation (Schulman et al., 2015; Lillicrap et al., 2016; Duan et al., 2016; Schulman et al., 2017; Fujimoto et al., 2018; Haarnoja et al., 2018). Despite promising empirical results, these widely applicable continuous RL algorithms execute a newly computed action at every step of the finest time scale of a problem. With no decision making at higher levels, they attempt to solve the challenging credit assignment problem over a long horizon. As a result, considerable sample efficiency improvements have yet to be made by them in complex task structures (Riedmiller et al., 2018; Li et al., 2020; Lee et al., 2020b) and extremely sparse reward settings (Andrychowicz et al., 2017; Plappert et al., 2018; Zhang et al., 2021).

On the other hand, it is argued that temporal abstraction (Parr and Russell, 1998; Dietterich, 1998; Sutton et al., 1999; Precup, 2000) is one of the crucial keys to solving control problems with complex structures. Larger steps are taken at higher levels of abstraction while lower-level actions only need to focus on solving isolated subtasks (Dayan and Hinton, 1993; Vezhnevets et al., 2017; Bacon et al., 2017). However, most hierarchical RL (HRL) methods are task specific and nontrivial to adapt. For example, the options framework (Sutton et al., 1999; Precup, 2000; Bacon et al., 2017) requires pre-defining an option space, while the feudal RL framework Vezhnevets et al. (2017); Nachum et al. (2018); Zhang et al. (2021) requires tuning the hyperparameters of dimensionality and domain range of the goal space. In practice, their final performance usually hinges on these choices.

Perhaps the simplest form of an option or sub-policy would be just repeating an action for a certain number of steps, a straightforward idea that has been widely explored (Lakshminarayanan et al.,

2017; Sharma et al., 2017; Dabney et al., 2021; Metelli et al., 2020; Lee et al., 2020a; Biedenkapp et al., 2021). This line of works can be regarded as a middle ground between "flat" RL and HRL. They assume a fixed candidate set of action durations, and repeat actions in an *open-loop* manner. Open-loop control forces an agent to commit to the same action over a predicted duration with no opportunity of early terminations. It weakens the agent's ability of handling emergency situations and correcting wrong durations predicted earlier. To address this inflexibility, a handful of prior works (Neunert et al., 2020; Chen et al., 2021) propose to output an "act-or-repeat" binary decision to decide if the action at the previous step should be repeated. Because this "act-or-repeat" decision will be examined at every step depending on the current environment state, this results in *closed-loop* action repetition.

All these action-repetition methods are well justified by the need of action *persistence* (Dabney et al., 2021; Amin et al., 2021; Zhang and Van Hoof, 2021; Grigsby et al., 2021) for designing a good exploration strategy, when action *diversity* should be traded for it properly. This trade-off is important because when reward is sparse or short-term reward is deceptive, action diversity alone only makes the agent wandering around its local neighborhood since any persistent trajectory has an exponentially small probability. In such a case, a sub-optimal solution is likely to be found. In contrast, persistence via action repetition makes the policy explore deeper (while sacrificing action diversity to some extent).

This paper further explores in the direction of closed-loop action repetition, striving to discover a novel algorithm that instantiates this idea better. The key question we ask is, how can we exploit the special structure of closed-loop repetition, so that our algorithm yields better sample efficiency and final performance compared to existing methods? As an answer to this question, we propose **t**emporally **a**bstract **a**ctor-**c**ritic (TAAC), a simple but effective off-policy RL algorithm that incorporates closed-loop action repetition into an actor-critic framework. Generally, we add a second stage that chooses between a candidate action output by an actor and the action from the previous step (Figure 1). Crucially, its "act-or-repeat" decision hinges on the actually sampled individual action instead of the expected behavior of the actor unlike recent works (Neunert et al., 2020; Chen et al., 2021). This post-acting switching scheme let the overall policy make more informed decisions. Moreover,

i) for policy evaluation, we propose a new compare-through Q operator for multi-step TD backup tailored to the action repetition scenario, instead of replying on generic importance correction;

ii) for policy improvement, we compute the actor gradient by multiplying a scaling factor to the $\frac{\partial Q}{\partial a}$ term from DDPG (Lillicrap et al., 2016) and SAC (Haarnoja et al., 2018), where the scaling factor is the optimal probability of choosing the actor's action in the second stage.

TAAC is much easier to train compared to sophisticated HRL methods, while it has two important features compared to "flat" RL algorithms, namely, persistent exploration and native multi-step TD backup support without the need of off-policyness correction.

We evaluate TAAC on 14 continuous control tasks, covering simple control, locomotion, terrain walking (Brockman et al., 2016), manipulation (Plappert et al., 2018), and self-driving (Dosovitskiy et al., 2017). Averaged over these tasks, TAAC largely outperforms 6 strong baselines. Importantly, our results show that it is our concrete instantiation of closed-loop action repetition that is vital to the final performance. The mere idea of repeating actions in a closed-loop manner doesn't guarantee better results than the compared open-loop methods. Moreover, our surprising finding reveals that while achieving top performance, TAAC is able to "mine" a significant number of repeated actions with the trained policy even on continuous tasks whose problem structures on the surface seem to repel action repetition (Section 5.6.2). This suggests that aside from encouraging persistent exploration, action repetition can find its place in a good policy behavior. This is perhaps due to that the action frequency of a task can be difficult to be set exactly as the minimum value that doesn't comprise optimal control while leaving no room for temporal abstraction (Grigsby et al., 2021).

## 2 Related work

Under the category of temporal abstraction via action repetition, there have been various formulations. Dabney et al. (2021) proposes temporally extended $\epsilon$-greedy exploration where a duration for repeating actions is sampled from a pre-defined truncated zeta distribution. This strategy only affects the exploration behavior for generating off-policy data but does not change the training objective. Sharma et al. (2017) and Biedenkapp et al. (2021) learn a hybrid action space and treat the discrete

action as a latent variable of action repeating steps, but the introduced temporal abstraction is open-loop and lacks flexibility. One recent work close to TAAC is PIC (Chen et al., 2021) which also learns to repeat the last action to address the action oscillation issue within consecutive steps. However, PIC was proposed for discrete control and its extension to continuous control is unclear yet. Also, PIC predicts whether to repeat the last action *independent of* a newly sampled action, which requires its switching policy to make a decision regarding the core policy's *expected* behavior. In an application section, H-MPO (Neunert et al., 2020) explored how continuous control can benefit from a meta binary action that modifies the overall system behavior. Again, like PIC their binary decision is made *in parallel with* a newly sampled action. Different from PIC and H-MPO, TAAC only decides "act-or-repeat" after comparing the previous action with a newly sampled action. Moreover, TAAC employs a new compare-through Q operator to exploit repeated actions for multi-step TD backup, and is trained by a much simpler actor gradient by absorbing the closed-form solution of the binary policy into the continuous action objective to avoid parameterizing a discrete policy unlike H-MPO.

Our experiment design (Section 5.2) has covered most methods that consider action repetition. Table 1 provides a checklist of the differences between TAAC and these methods.

# 3 Preliminaries

We consider the RL problem as policy search in a Markov Decision Process (MDP). Let $s \in \mathbb{R}^M$ denote a state, where a continuous action $a \in \mathbb{R}^N$ is taken. Let $\pi(a|s)$ be the action policy, and $\mathcal{P}(s_{t+1}|s_t, a_t)$ the probability of the environment transitioning to $s_{t+1}$ after an action $a_t$ is taken at $s_t$. Upon reaching $s_{t+1}$, the agent receives a scalar reward $r(s_t, a_t, s_{t+1})$. The RL objective is to find a policy $\pi^*$ that maximizes the expected discounted return: $\mathbb{E}_{\pi,\mathcal{P}}\left[\sum_{t=0}^\infty \gamma^t r(s_t, a_t, s_{t+1})\right]$, where $\gamma \in (0, 1)$ is a discount factor. We also define $Q^\pi(s_t, a_t) = \mathbb{E}_\pi\left[\sum_{t'=t}^\infty \gamma^{t'-t} r(s_{t'}, a_{t'}, s_{t'+1})\right]$ as the discounted return starting from $s_t$ given that $a_t$ is taken and then $\pi$ is followed, and $V^\pi(s_t) = \mathbb{E}_{a_t \sim \pi} Q^\pi(s_t, a_t)$ as the discounted return starting from $s_t$ following $\pi$.

In an off-policy actor-critic setting with $\pi$ and $Q$ parameterized by $\phi$ and $\theta$, a surrogate objective is usually used (Lillicrap et al., 2016; Haarnoja et al., 2018)

$$\max_\phi \mathbb{E}_{s \sim \mathcal{D}} V_\theta^{\pi_\phi}(s) \triangleq \max_\phi \mathbb{E}_{s \sim \mathcal{D}, a \sim \pi_\phi} Q_\theta(s, a). \tag{1}$$

This objective maximizes the expected state value over some state distribution, assuming that 1) $s$ is sampled from a replay buffer $\mathcal{D}$ instead of the current policy, and 2) the dependency of the critic $Q_\theta(s, a)$ on the policy $\pi_\phi$ is dropped when computing the gradient of $\phi$. Meanwhile, $\theta$ is learned separately via policy evaluation with typical TD backup.

# 4 Temporally abstract actor-critic

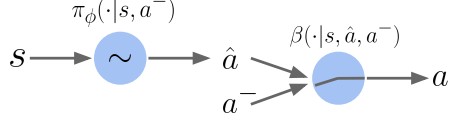

To enable temporal abstraction, we decompose the agent's action decision into two stages (Figure 1): 1) sampling a new candidate action $\hat{a} \sim \pi_\phi(\cdot|s, a^-)$ conditioned on the action $a^-$ at the previous time step, and 2) choosing between $a^-$ and $\hat{a}$ as the actual output at the current step. The overall TAAC algorithm is summarized in Algorithm 1 Appendix A.

Figure 1: TAAC's two-stage policy during inference. In the first stage, an action policy $\pi_\phi$ samples a candidate action $\hat{a}$. In the second stage, a binary switching policy $\beta$ chooses between this candidate and the previous action $a^-$.

## 4.1 Two-stage policy

Formally, let $\beta(b|s, \hat{a}, a^-)$ be the binary switching policy, where $b = 0/1$ means choosing $a^-/\hat{a}$. For simplicity, in the following we will denote $\beta_b = \beta(b|s, \hat{a}, a^-)$ (always assuming its dependency on $s$, $\hat{a}$, and $a^-$). Then our two-stage policy $\pi^{\text{ta}}$ for temporal abstraction is defined as

$$\pi^{\text{ta}}(a|s, a^-) \triangleq \int_{\hat{a}} \pi_\phi(\hat{a}|s, a^-) \left[\beta_0 \delta(a - a^-) + \beta_1 \delta(a - \hat{a})\right] d\hat{a}, \tag{2}$$

which can be shown to be a proper probability distribution of $a$. This two-stage policy repeats previous actions through a binary policy $\beta$, a decision maker that compares $a^-$ and $\hat{a}$ side by side

given the current state $s$. Repeatedly favoring $b = 0$ results in temporal abstraction of executing the same action for multiple steps. Moreover, this control is closed-loop, as it does not commit to a pre-determined time window; instead it can stop repetition whenever necessary. As a special case, when $\beta_1 = 1$, $\pi^{\text{ta}}$ reduces to $\pi_\phi$; when $\beta_0 = 1$, $\pi^{\text{ta}}(a|s, a^-) = \delta(a - a^-)$.

## 4.2 Policy evaluation with the compare-through operator

The typical one-step TD learning objective for a policy $\pi$ is

$$\min_\theta \mathbb{E}_{(s,a,s')\sim\mathcal{D}} [Q_\theta(s,a) - \mathcal{B}^\pi Q_{\bar\theta}(s,a)]^2 \text{, with } \mathcal{B}^\pi Q_{\bar\theta}(s,a) = r(s,a,s') + \gamma V_{\bar\theta}^\pi(s'), \quad (3)$$

where $\mathcal{B}^\pi$ is the Bellman operator, and $\bar\theta$ slowly tracks $\theta$ to stabilize the learning (Mnih et al., 2015). For multi-step bootstrapping, importance correction is usually needed, for example, Retrace (Munos et al., 2016) relies on a function of $\frac{\pi(a|s)}{\mu(a|s)}$ ($\mu$ is a behavior policy) to correct for the off-policyness of a trajectory. Unfortunately, our $\pi^{\text{ta}}$ makes probability density computation challenging because of the marginalization over $\hat{a}$. Thus importance correction methods including Retrace cannot be applied to our case. To address this issue, below we propose a new multi-step Q operator, called *compare-through*. Then we explain how $\pi^{\text{ta}}$ can exploit this operator for efficient policy evaluation.

For a learning policy $\pi$, given a trajectory $(s_0, a_0, s_1, a_1, \ldots, s_N, a_N)$ from a behavior policy $\mu$, let $\tilde{a}_n$ denote the actions sampled from $\pi$ at states $s_n (n \geq 1)$ and $\tilde{a}_0 = a_0$ (we do not sample from $\pi$ at $s_0$). We define (a point estimate of) the compare-through operator $\mathcal{T}^\pi$ as

$$\mathcal{T}^\pi Q_{\bar\theta}(s_0, a_0) \approx \sum_{t=0}^{n-1} \gamma^t r(s_t, a_t, s_{t+1}) + \gamma^n Q_{\bar\theta}(s_n, \tilde{a}_n), \quad (4)$$

where $n = \min(\{n : \tilde{a}_n \neq a_n\} \cup \{N\})$. Intuitively, given a sampled trajectory of length $N$ from the replay buffer, the compare-through operator takes an expectation, under the current policy at the sampled states (from $s_1$ to $s_N$), over all the sub-trajectoriess (up to length $N$) of actions that match the sampled actions. Note that Eq. 4 is a *point estimate* of this expectation. A formal definition of $\mathcal{T}^\pi$ is described by Eq. 17, and its relation to Retrace (Munos et al., 2016) is shown in Appendix L.

**Theorem 1 (Policy evaluation convergence)** *In a tabular setting, the compare-through operator $\mathcal{T}^\pi$, whose point estimate defined by Eq. 4 (without the parameters $\bar\theta$) and expectation form defined by Eq. 17, has a unique fixed point $Q^\pi$, where $\pi$ is the current (target) policy.*

For the detailed proof we refer the reader to Appendix L. Although the actual setting considered in this paper are continuous state and action domains with function approximators, Theorem 1 still provides some justification for our compare-through operator.

Clearly, any discrete policy could exploit the compare-through operator since there can be a non-zero chance of two discrete actions being compared equal. A typical stochastic continuous policy such as Gaussian used by SAC (Haarnoja et al., 2018) always has $\tilde{a}_n \neq a_n$ for $n \geq 1$ w.r.t. any behavior policy $\mu$. In this case $\mathcal{T}^\pi$ is no more than just a Bellman operator $\mathcal{B}^\pi$. However, if a continuous policy is specially structured to be "action-reproducible", it will enjoy the privilege of using $s_n$ for $n > 1$ as the target state. Our two-stage $\pi^{\text{ta}}$ is such a case, where each action $a_n$ ($\tilde{a}_n$) is accompanied by a repeating choice $b_n$ ($\tilde{b}_n$). Starting from $n = 1$ with a previous action $a_0$, if $\tilde{b}_m = b_m = 0$ (both repeated) for all $1 \leq m \leq n$, then we know that $\tilde{a}_n = a_n = \ldots = \tilde{a}_1 = a_1 = a_0$. In other words, if two trajectories start with the same

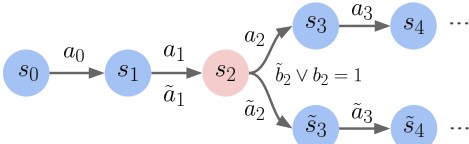

Figure 2: An illustration of the compare-through operator by exploiting action repetition of $\pi^{\text{ta}}$. The upper branch is the trajectory sampled by a rollout policy; the lower one is sampled by the current policy during training. We have $\tilde{a}_1 = a_1 = a_0$ due to $b_1 = \tilde{b}_1 = 0$. The two trajectories diverge at $s_2$ because either $b_2 = 1$ or $\tilde{b}_2 = 1$. For bootstrapping, we use $s_2$ as the target state in this example.

$(s_0, a_0)$, we can compare their discrete $\{b_n\}$ sequences in place of the continuous $\{a_n\}$ sequences. See Figure 2 for an illustration. Thus for multi-step TD learning we use $\mathcal{T}^{\pi^{\text{ta}}}$ to replace $\mathcal{B}^\pi$ in Eq. 3.

**Remark** The compare-through operator is not meant to replace multi-step TD learning with importance correction in a general scenario, as it is only effective for "action-reproducible" policies. Its bootstrapping has a hard cutoff (by checking $\tilde{a}_n \neq a_n$) instead of a soft one as in Munos et al. (2016).

**On the importance of reward normalization** Because $\mathcal{T}^{\pi^{\text{ta}}}$ computes bootstrapping targets based on dynamic step lengths and rewards are propagated faster with greater lengths, Q values bootstrapped by greater lengths might become overly optimistic/pessimistic (*e.g.*, imagine a task with rewards that are all positive/negative). This could affect the policy selecting actions according to their Q values. However, this side effect is only *temporary* and will vanish if all Q values are well learned eventually. In practice, we find it beneficial to normalize the immediate reward by its moving average statistics to roughly maintain a zero mean and unit standard deviation (detailed in Appendix I.1).

### 4.3   Policy improvement with a closed-form $\beta^*$

It can be shown (Appendix B) that the parameterized state value of $\pi^{\text{ta}}$ is

$$V_\theta^{\pi^{\text{ta}}}(s|a^-) = \mathop{\mathbb{E}}_{\hat{a}\sim\pi_\phi, b\sim\beta} \left[ (1-b)Q_\theta(s, a^-) + bQ_\theta(s, \hat{a}) \right], \tag{5}$$

Intuitively, the actual Q value of each $a \sim \pi^{\text{ta}}$ is an interpolation by $\beta$ between the Q values of $a^-$ and $\hat{a}$. Our policy improvement objective is then $\max_{\phi,\beta} \mathbb{E}_{(s,a^-)\sim\mathcal{D}} V_\theta^{\pi^{\text{ta}}}(s|a^-)$. Note that each time we sample the previous action $a^-$ along with a state $s$ from the replay buffer. To encourage exploration, following prior works (Mnih et al., 2016; Riedmiller et al., 2018) we augment this objective with a joint entropy $\mathcal{H}_{s,a^-} = \mathbb{E}_{\pi_\phi(\hat{a}|s,a^-)\beta_b} \left[ -\log\beta_b - \log\pi_\phi(\hat{a}|s,a^-) \right]$. Thus the final policy improvement objective is

$$\begin{aligned}
&\max_{\phi,\beta} \mathop{\mathbb{E}}_{(s,a^-)\sim\mathcal{D}} \left[ V_\theta^{\pi^{\text{ta}}}(s|a^-) + \alpha\mathcal{H}_{s,a^-} \right] \\
&= \max_{\phi,\beta} \mathop{\mathbb{E}}_{\substack{(s,a^-)\sim\mathcal{D} \\ \hat{a}\sim\pi_\phi, b\sim\beta}} \left[ (1-b)Q_\theta(s, a^-) + bQ_\theta(s, \hat{a}) - \alpha\left(\log\beta_b + \log\pi_\phi(\hat{a}|s,a^-)\right) \right],
\end{aligned} \tag{6}$$

where $\alpha$ is a temperature parameter. Given any $(s, a^-, \hat{a})$, we can derive a closed-form solution of the (non-parametric) $\beta$ policy for the innermost expectation $b \sim \beta$ as

$$\beta_1^* = \exp\left(\frac{Q_\theta(s,\hat{a})}{\alpha}\right) \Big/ \left( \exp\left(\frac{Q_\theta(s,\hat{a})}{\alpha}\right) + \exp\left(\frac{Q_\theta(s,a^-)}{\alpha}\right) \right).$$

Then applying the re-parameterization trick $\hat{a} = f_\phi(\epsilon, s, a^-), \epsilon \sim \mathcal{N}(0, I)$, one can show that the estimated actor gradient is

$$\begin{aligned}
\Delta\phi &\triangleq \left( \beta_1^* \frac{\partial Q_\theta(s, f_\phi)}{\partial\phi} - \alpha\frac{\partial\log\pi_\phi(f_\phi|s,a^-)}{\partial\phi} \right) \\
&= \left( \beta_1^* \frac{\partial Q_\theta(s, \hat{a})}{\partial\hat{a}} - \alpha\frac{\partial\log\pi_\phi(\hat{a}|s,a^-)}{\partial\hat{a}} \right) \frac{\partial f_\phi}{\partial\phi} - \alpha\frac{\partial\log\pi_\phi(\hat{a}|s,a^-)}{\partial\phi}.
\end{aligned} \tag{7}$$

This gradient has a very similar form with SAC's (Haarnoja et al., 2018), except that here $\frac{\partial Q}{\partial\hat{a}}$ has a scaling factor $\beta_1^*$. We refer the reader to a full derivation of the actor gradient in Appendix D.

**Remark on $\beta^*$** This closed-form solution is only possible after $\hat{a}$ is sampled. It's essentially comparing the Q values between $a^-$ and $\hat{a}$. This side-by-side comparison is absent in previous closed-loop repeating methods like PIC (Chen et al., 2021) and H-MPO (Neunert et al., 2020).

**Remark on multi-step actor gradient** According to Figure 1, the newly generated action $\hat{a}_t$ at the current step $t$, if repeated as future $a_{t+1}^-, \ldots, a_{t'}^-$ ($t' > t$), will also influence the maximization of future V values: $V_\theta^{\pi^{\text{ta}}}(s_{t+1}|a_{t+1}^-), \ldots, V_\theta^{\pi^{\text{ta}}}(s_{t'}|a_{t'}^-)$. As a result, in principle $\hat{a}_t$ has a multi-step gradient. To exactly compute this multi-step gradient, a fresh rollout of the current $\pi^{\text{ta}}$ via interacting with the environment is necessary. For reasons detailed in Appendix E, we end up truncating this full gradient to the first step, by letting Eq. 6 sample $(s, a^-)$ pairs at non-consecutive time steps from the replay buffer. This one-step truncation is also (implicitly) adopted by H-MPO (Neunert et al., 2020) for their action repetition application case. Interestingly, the truncation results in a simple implementation of actor gradient of TAAC similar to SAC's.

**Automatically tuned temperatures** Given two entropy targets $\mathcal{H}'$ and $\mathcal{H}''$, we learn temperatures $\alpha'$ and $\alpha''$ from the objective

$$\min_{\substack{\log(\alpha'), \\ \log(\alpha'')}} \left\{ \mathop{\mathbb{E}}_{(s,a^-)\sim\mathcal{D}, \hat{a}\sim\pi_\phi, b\sim\beta^*} \left[ \log(\alpha')(-\log\beta_b^* - \mathcal{H}') + \log(\alpha'')(-\log\pi_\phi(\hat{a}|s,a^-) - \mathcal{H}'') \right] \right\}, \tag{8}$$

| | SAC (Haarnoja et al., 2018) | SAC-Ntd | SAC-Nrep | SAC-Krep (Sharma et al., 2017) (Biedenkapp et al., 2021) | SAC-EZ (Dabney et al., 2021) | SAC-Hybrid (Neunert et al., 2020) | TAAC-1td | TAAC |
|---|---|---|---|---|---|---|---|---|
| Persistent exploration | ✗ | ✗ | ✓ | ✓ | ✓ | ✓ | ✓ | ✓ |
| Multi-step TD | ✗ | ✓ | ✓ | ✓ | ✗ | ✓ | ✗ | ✓ |
| Closed-loop repetition | ✗ | ✗ | ✗ | ✗ | ✗ | ✓ | ✓ | ✓ |
| Learnable duration | ✗ | ✗ | ✗ | ✓ | ✗ | ✓ | ✓ | ✓ |

Table 1: A summary of the 8 major comparison methods in our experiments.

by adjusting $\log(\alpha)$ instead of $\alpha$ as in SAC (Haarnoja et al., 2018). We learn temperatures for $\pi_\phi$ and $\beta$ separately, to enable a finer control of the two policies and their entropy terms, similar to the separate policy constraints (Abdolmaleki et al., 2018; Neunert et al., 2020). Appendix F shows how several equations slightly change if two temperatures are used ($\alpha$ is replaced by $\alpha'$ or $\alpha''$).

# 5 Experiments

## 5.1 Tasks

To test if the proposed algorithm is robust and can be readily applied to many tasks, we perform experiments over 14 continuous control tasks under different scenarios:

a) **SimpleControl**: Three control tasks (Brockman et al., 2016) with small action and observation spaces: *MountainCarContinuous*, *LunarLanderContinuous*, and *InvertedDoublePendulum* ;
b) **Locomotion**: Four locomotion tasks (Brockman et al., 2016) that feature complex physics and action spaces: *Hopper*, *Ant*, *Walker2d*, and *HalfCheetah*;
c) **Terrain**: Two locomotion tasks that require adapting to randomly generated terrains: *BipedalWalker* and *BipedalWalkerHardcore*;
d) **Manipulation**: Four Fetch (Plappert et al., 2018) tasks with sparse rewards and hard exploration (reward given only upon success): *FetchReach*, *FetchPush*, *FetchSlide*, and *FetchPickAndPlace*;
e) **Driving**: One CARLA autonomous-driving task (Dosovitskiy et al., 2017) that has complex high-dimensional multi-modal sensor inputs (camera, radar, IMU, collision, GPS, *etc.*): *Town01*. The goal is to reach a destination starting from a randomly spawned location in a small realistic town, while avoiding collisions and red light violations.

Among the 5 categories, (d) and (e) might benefit greatly from temporal abstraction because of hard exploration or the problem structure (*e.g.*, driving naturally involves repeated actions). Categories (a)-(c) appear unrelated with temporal abstraction, but we test if seemingly unrelated tasks can also benefit from it. By comparing TAAC against different methods across vastly different tasks, we hope to demonstrate its generality, because adaptation to this kind of task variety requires few assumptions about the task structure and inputs/outputs. For more task details, we refer the reader to Appendix I.

## 5.2 Comparison methods

While there exist many off-policy hierarchical RL methods that model temporal abstraction, for example Nachum et al. (2018); Riedmiller et al. (2018); Levy et al. (2019); Li et al. (2020); Zhang et al. (2021), we did not find them readily scalable to our entire list of tasks (especially to high dimensional input space like CARLA), without considerable efforts of adaptation. Thus our primary focus is to compare TAAC with baselines of different formulations of action repetition: vanilla SAC (Haarnoja et al., 2018), SAC-Nrep, SAC-Krep (Sharma et al., 2017; Biedenkapp et al., 2021), SAC-EZ (Dabney et al., 2021), and SAC-Hybrid (Neunert et al., 2020). Although originally some baselines have their own RL algorithm backbones, in this experiment we choose SAC as the common backbone because: 1) SAC is state-of-the-art among open-sourced actor-critic algorithms, 2) a common backbone facilitates reusing the same set of core hyperparameters for a fairer comparison, and 3) by reducing experimental variables, it gives us a better focus on the design choices of action repetition instead of other orthogonal algorithmic components.

Let $N$ be a parameter controlling the maximal number of action repeating steps. SAC-Nrep simply repeats every action $N$ times. SAC-Krep, inspired by FiGAR (Sharma et al., 2017) and TempoRL (Biedenkapp et al., 2021), upgrades an action $a$ to a pair of $(a, K)$, indicating that the agent will repeat action $a$ for the next $K$ steps ($1 \le K \le N$) without being interrupted until finishing. To implement SAC-Krep, following Delalleau et al. (2020) we extended the original SAC algorithm to support a mixture of continuous and discrete actions (Appendix G). Note that SAC-Krep's open-loop

control is in contrast to TAAC's closed-loop control. SAC-EZ incorporates the temporally extended $\epsilon$-greedy exploration (Dabney et al., 2021) into SAC. During rollout, if the agent decides to explore, then the action is uniformly sampled and the duration for repeating that action is sampled from a truncated zeta distribution $zeta(n) \propto n^{-\mu}$, $1 \le n \le N$. This fixed duration model encourages persistent exploration depending on the value of the hyperparameter $\mu$. The training step of SAC-EZ is the same with SAC. SAC-Hybrid shares a similar flavor of H-MPO (Neunert et al., 2020) for closed-loop action repetition. It defines a hybrid policy to output the continuous action and binary switching action in parallel, assuming their conditional independence given the state. This independence between hybrid actions and how the Q values are computed in SAC-Hybrid are the biggest differences with TAAC. We also apply Retrace (Munos et al., 2016) to its policy evaluation with $N$-step TD as done by H-MPO. We refer the reader to the algorithm details of SAC-Hybrid in Appendix H.

In order to analyze the benefit of persistent exploration independent of that of multi-step TD learning, we also compare two additional methods. SAC-Ntd is a variant of SAC where a Q value is bootstrapped by an $N$-step value target with Retrace (Munos et al., 2016) to correct for off-policyness. For ablating TAAC, we evaluate TAAC-1td that employs a typical Bellman operator $\mathcal{B}^{\pi^{\text{ta}}}$ for one-step bootstrapping. Thus we have 8 methods in total for comparison in each task. See Table 1 for a summary and Appendix J for method details.

In our experiments, we set the repeating hyperparameter $N$ to 3 on **SimpleControl**, **Locomotion** and **Manipulation**, and to 5 on **Terrain** and **Driving**. Here the consideration of $N$ value is mainly for open-loop methods like SAC-Nrep and SAC-Krep because they will yield poor performance with large values of $N$. TAAC is not sensitive to $N$ for policy evaluation, as shown in Section 5.5.

## 5.3 Evaluation protocol

To measure the final model quality, we define *score* as the episodic return $\sum_{t=0}^{T} r(s_t, a_t, s_{t+1})$ of evaluating (by taking the approximate mode of a parameterized continuous policy; see Appendix C for details) a method for a task episode, where $T$ is a pre-defined time limit or when the episode terminates early. Different tasks, even within the same group, can have vastly different reward scales (*e.g.*, tens *vs.* thousands). So it is impractical to directly average their scores. It is not uncommon in prior works (Hessel et al., 2018) to set a performance range for each task separately, and normalize the score of that task to roughly $[0, 1]$ before averaging scores of a method over multiple tasks. Similarly, to facilitate score aggregation across tasks, we adopt the metric of *normalized score* (*n-score*). For each task, we obtain the score (averaged over 100 episodes) of a random policy and denote it by $Z_0$. We also evaluate the best method on that task and obtain its average score $Z_1$. Given a score $Z$, its normalized value is calculated as $\frac{Z-Z_0}{Z_1-Z_0}$. With this definition, the n-score of each task category (a)-(e) can be computed as the averaged n-score across tasks within that category. Additionally, to measure training convergence speed, we approximate *n-AUC* (area under the n-score curve normalized by $x$ value range) by

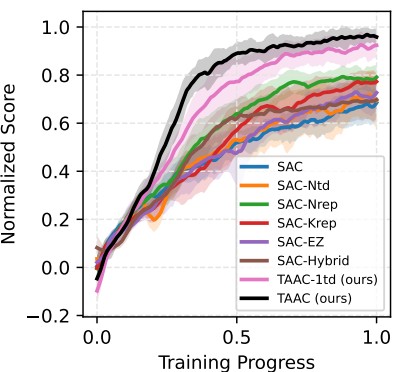

Figure 3: Training curves (n-score *vs.*training progress) of the 8 comparison methods in one plot. Each curve is a mean of a method's n-score curves over all the 14 tasks, where the method is run with 3 random seeds on each task. See Figure 8 and Figure 9 (Appendix K) for the complete set of individual training curves.

averaging n-scores on a n-score curve throughout the training. A higher n-AUC value indicates a faster convergence speed. n-AUC is a secondary metric to look at when two methods have similar final n-scores. Finally, by averaging n-score and n-AUC values over multiple tasks, we emphasize the robustness of an RL method.

Given a task, we train each method for the same number of environment frames. Crucially, for fair comparisons we also make each method train 1) for the same number of gradient steps, 2) with the same mini-batch size and learning rate, 3) using roughly the same number of weights, and 4) with a common set of hyperparameters (tuned with vanilla SAC) for the SAC backbone . More details of the experimental settings are described in Appendix J.

| | | SAC | SAC-Ntd | SAC-Nrep | SAC-Krep | SAC-EZ | SAC-Hybrid | TAAC-1td | TAAC |
|---|---|---|---|---|---|---|---|---|---|
| | **SimpleControl** | 0.64[0.03] | 0.54[0.07] | 0.84[0.01] | 0.55[0.03] | 0.65[0.03] | 0.88[0.14] | 0.83[0.13] | **0.99**[0.04] |
| | **Locomotion** | 0.89[0.06] | 0.88[0.11] | 0.70[0.08] | 0.82[0.07] | **0.92**[0.07] | 0.63[0.08] | 0.88[0.04] | 0.90[0.05] |
| Final n-score | **Terrain** | 0.35[0.10] | 0.48[0.02] | 0.54[0.03] | 0.48[0.03] | 0.45[0.15] | 0.79[0.03] | 0.96[0.02] | **1.00**[0.03] |
| (model quality) | **Manipulation** | 0.60[0.12] | 0.75[0.17] | 0.91[0.07] | 0.98[0.05] | 0.68[0.07] | 0.49[0.07] | **0.99**[0.02] | **0.99**[0.01] |
| | **Driving** | 0.88[0.05] | 0.84[0.04] | 0.92[0.03] | 0.95[0.04] | 0.71[0.03] | **1.00**[0.04] | 0.92[0.02] | 0.97[0.04] |
| | **All** | 0.68[0.08] | 0.71[0.10] | 0.78[0.05] | 0.77[0.05] | 0.71[0.07] | 0.69[0.08] | 0.92[0.05] | **0.96**[0.03] |
| | **SimpleControl** | 0.45[0.01] | 0.41[0.01] | 0.51[0.02] | 0.28[0.03] | 0.45[0.02] | 0.62[0.08] | 0.60[0.10] | **0.72**[0.04] |
| | **Locomotion** | 0.69[0.03] | 0.64[0.07] | 0.55[0.05] | 0.59[0.04] | 0.64[0.05] | 0.50[0.06] | 0.72[0.02] | **0.74**[0.05] |
| n-AUC | **Terrain** | 0.17[0.02] | 0.19[0.02] | 0.38[0.01] | 0.23[0.01] | 0.21[0.03] | 0.50[0.01] | 0.50[0.02] | **0.59**[0.02] |
| (convergence | **Manipulation** | 0.41[0.04] | 0.50[0.10] | 0.69[0.06] | 0.71[0.08] | 0.49[0.06] | 0.38[0.02] | 0.71[0.04] | **0.77**[0.02] |
| speed) | **Driving** | 0.38[0.02] | 0.41[0.04] | 0.52[0.02] | 0.51[0.03] | 0.32[0.02] | 0.61[0.02] | 0.60[0.03] | **0.65**[0.02] |
| | **All** | 0.46[0.03] | 0.47[0.06] | 0.55[0.04] | 0.50[0.05] | 0.47[0.04] | 0.50[0.04] | 0.65[0.04] | **0.72**[0.03] |

Table 2: n-score and n-AUC results. Margins in brackets are computed by averaging the standard deviations (across 3 random seeds) of individual tasks. The last two shaded columns are our methods.

## 5.4 Results and observations

The n-AUC and final n-score values are shown in Table 2, and the training curves of n-score are shown in Figure 3. First of all, we conclude that the tasks are diverse enough to reduce the evaluation variance. The averaged standard deviations are small for most methods. Thus the comparison results are not coincidences and are likely to generalize to other scenarios. Overall, TAAC largely outperforms the 6 baselines regarding both final n-score (0.96[0.03] *vs.* second-best 0.78[0.05]) and n-AUC (0.72[0.03] *vs.* second-best 0.55[0.04]), with relatively small standard deviations. Note that the n-AUC gap naturally tends to be smaller than the final n-score gap because the former reflects a convergence trend throughout the training. Moreover, TAAC achieved top performance of n-AUC on each individual task category. Although some baselines achieved best final n-scores by slightly dominating TAAC, their performance is not consistent over all task categories. More observations can be made below.

- *Persistent exploration and the compare-through operator are both crucial.* Even with one-step TD, TAAC-1td's performance (0.92[0.05] and 0.65[0.04]) already outperforms the baselines. This shows that persistent exploration alone helps much. Furthermore, TAAC is generally better than TAAC-1td (0.96[0.03] *vs.* 0.92[0.05] and 0.72[0.03] *vs.* 0.65[0.04]). This shows that efficient policy evaluation by the compare-through operator is also a key component of TAAC.
- *A proper formulation of closed-loop action repetition is important.* The idea of closed-loop action repetition alone is not a magic ingredient, as SAC-Hybrid only has moderate performance among the baselines. Notably, it performs worst on **Locomotion** and **Manipulation**. Our analysis revealed that its "act-or-repeat" policy tends to repeat with very high probabilities on *Ant*, *FetchPickAndPlace*, and *FetchPush* even towards the end of training, resulting in very poor n-scores. All these three tasks feature hard exploration. This result suggests that a good formulation of the idea is crucial. The two-stage decision $\pi(\hat{a}|s, a^-)\pi(b|s, a^-, \hat{a})$ of TAAC is clearly more informed than the parallel decisions $\pi(\hat{a}|s, a^-)\pi(b|s, a^-)$ of SAC-Hybrid. Furthermore, even with a latent "act-or-repeat" action, TAAC manages to maintain the complexity of the Q function $Q(s, a)$ as in the original control problem, while SAC-Hybrid has a more complex form $Q((s, a^-), (\hat{a}, b))$ (Appendix H).
- *Naive action repetition works well.* Interestingly, in this particular experiment, SAC-Nrep is the overall top-performing baseline due to its relatively balanced results on all task categories. However, when it fails, the performance could be very bad (**Locomotion** and **Terrain**). While its sample efficiency is good, it has a *difficulty of approaching the optimal control* (final mean n-scores $\leq 0.92$ in individual task categories) due to its lack of flexibility in the action repeating duration. This suggests that action repetition greatly helps, but a fixed duration is difficult to pick.
- *Limitation: action repetition hardly benefits tasks with densely shaped rewards and frame skipping.* We notice that TAAC is no better than SAC and SAC-EZ on **Locomotion** regarding the final n-score, although it has slight advantages of n-AUC. The locomotion tasks have densely shaped rewards to guide policy search. Thus action repetition hardly helps locomotion exploration, especially when the 4 tasks already have built-in frameskips (4 frames for *Hopper* and *Walker2d*; 5 frames for *Ant* and *HalfCheetah*). We believe that more sophisticated temporal abstraction (*e.g.*, skills) is needed to improve the performance in this case.

## 5.5 Off-policyness experiments

To verify that our compare-through operator is not affected by off-policyness in policy evaluation, we compare TAAC to a variant TAAC-Ntd which always bootstraps a Q value with an $N$-step target,

| | Final n-score (model quality) | | | | | | n-AUC (convergence speed) | | | | | |
|---|---|---|---|---|---|---|---|---|---|---|---|---|
| | **S** | **L** | **T** | **M** | **D** | **All** | **S** | **L** | **T** | **M** | **D** | **All** |
| TAAC-Ntd ($N = 10$) | 0.88[0.16] | 0.73[0.18] | 0.79[0.06] | 0.82[0.07] | 0.72[0.01] | 0.79[0.12] | 0.56[0.11] | 0.54[0.12] | 0.53[0.03] | 0.62[0.04] | 0.38[0.06] | 0.55[0.08] |
| TAAC ($N = 10$) | 1.00[0.02] | 0.85[0.11] | 0.97[0.04] | 0.94[0.02] | 0.94[0.03] | 0.93[0.05] | 0.73[0.01] | 0.67[0.06] | 0.59[0.02] | 0.73[0.03] | 0.60[0.00] | 0.68[0.03] |
| SAC-Hybrid | 0.88[0.14] | 0.63[0.08] | 0.79[0.03] | 0.49[0.07] | 1.00[0.04] | 0.69[0.08] | 0.62[0.08] | 0.50[0.06] | 0.50[0.01] | 0.38[0.02] | 0.61[0.02] | 0.50[0.04] |
| SAC-Hybrid-CompThr | 0.89[0.12] | 0.65[0.08] | 0.75[0.03] | 0.68[0.17] | 0.90[0.04] | 0.74[0.11] | 0.58[0.08] | 0.48[0.06] | 0.47[0.01] | 0.47[0.09] | 0.49[0.01] | 0.50[0.06] |

Table 3: Off-policyness experiments results. Error margins inside brackets are computed by averaging the standard deviations (across 3 random seeds) of individual tasks in a category. **S**: **SimpleControl**; **L**: **Locomotion**; **T**: **Terrain**; **M**: **Manipulation**; **D**: **Driving**.

regardless of $\beta$'s outputs and without importance correction[1]. We choose a large trajectory length $N = 10$ to amplify the effect of off-policyness. Table 3 shows that $N$-step TD without importance correction significantly degrades the performance ($0.79[0.12]$ *vs.* $0.93[0.05]$ and $0.55[0.08]$ *vs.* $0.68[0.03]$). In contrast, the compare-through operator well addresses this issue for TAAC.

Furthermore, we implement a variant of SAC-Hybrid by replacing the Retrace operator with our compare-through operator, to see if the result difference between TAAC and SAC-Hybrid is mainly due to different ways of handling off-policyness. Table 3 shows that SAC-Hybrid-CompThr performs similarly to SAC-Hybrid ($0.74[0.11]$ *vs.* $0.69[0.08]$ and $0.50[0.06]$ *vs.* $0.50[0.04]$), suggesting that it is indeed the formulation of SAC-Hybrid that creates its performance gap with TAAC.

## 5.6 Policy behavior visualization and analysis

In this section, we mainly answer two questions:

1) How is the exploration behavior of TAAC compared to that of SAC, a "flat" RL algorithm?
2) What is the action repetition behavior of TAAC in a trained control task?

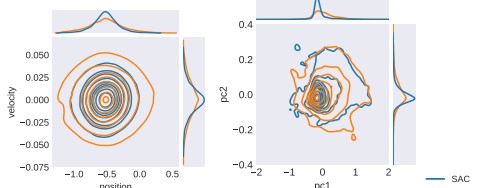

Figure 4: KDE plots of 100K state vectors visited by TAAC and SAC with their randomized policies. The side plots on top and right represent 1D marginal density functions. Left: *MountainCarContinuous*: $x$ is the car's position and $y$ is the car's velocity; Right: *BipedalWalker*: $xy$ represent the top-2 principal components of the walker's hull.

### 5.6.1 Exploration behavior

TAAC introduces persistent exploration along previous actions, having a better chance of escaping the local neighborhood around a state when acting randomly. Thus TAAC's exploration should yield a better state space coverage than SAC's does, assuming other identical conditions. To verify this, we visualize the state space coverage by SAC and TAAC during their initial exploration phases.

Specifically, we select two tasks *MountainCarContinuous* and *BipedalWalker* for this purpose. For either TAAC or SAC, we play a random version of its policy on either task for 50K environment frames, to simulate the initial exploration phase where the model parameters have not yet been altered by training. During this period, we record all the 50K state vectors for analysis. For *MountainCarContinuous*, each state vector is 2D, representing the car's "x-position" and "x-velocity" on the 1D track. For *BipedalWalker*, each state vector is 24D, where the first 4D sub-vector indicates the statistics of the walker's hull: "angle", "angular velocity", "x-velocity", and "y-velocity". For visualization, we first extract this 4D sub-vector and form a combined dataset of 100K vectors from both TAAC and SAC. Then we apply PCA (Jolliffe, 1986) to this dataset and project each 4D vector down to 2D. After this, we are able to draw KDE (kernel density estimate) plots for both *MountainCarContinuous* and *BipedalWalker* in Figure 4. We see that on both tasks, a random policy of TAAC is able to cover more diverse states compared to SAC. This suggests that in general, TAAC is better at exploration compared to a "flat" RL method, thanks to its ability of persistent exploration.

### 5.6.2 Action repetition behavior

In theory, to achieve optimal continuous control, the best policy should always sample a new action at every step and avoid action repetition at all. Thus one might assume that TAAC's second-stage policy shifts from frequently repeating actions in the beginning of training, to not repeating actions at all

---

[1]We cannot apply Retrace to TAAC-Ntd since the probability density of $\pi^{\text{ta}}$ is computationally intractable.

| Tasks | MCC | LLC | IDP | HOP | ANT | WAL | HC | BW | BWH | FR | FP | FS | FPP | TOW |
|---|---|---|---|---|---|---|---|---|---|---|---|---|---|---|
| **Action repetition percentage** | 89% | 74% | 26% | 37% | 13% | 26% | 1% | 25% | 39% | 9% | 55% | 54% | 49% | 55% |

Table 4: Action repetition percentages by evaluating a trained TAAC model for 100 episodes on each of the 14 tasks. Refer to Section 5.1 for the full task names.

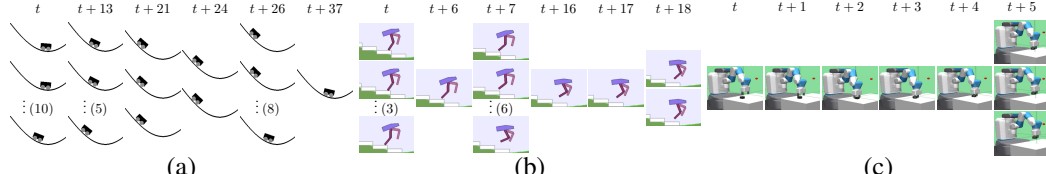

(a)          (b)          (c)

Figure 5: Example frames (cropped) from top-performing evaluation episodes of TAAC. Each column contains consecutive frame(s) generated by the same action, where "$\vdots (n)$" denotes $n$ similar frames omitted due to space limit. (a): In *MountainCarContinuous* the car first backs up to build gravitational potential and rushes down; (b): In *BipedalWalkerHardcore* the bipedal walker jumps one step down; (c): In *FetchPickAndPlace* the robot arm approaches the object and lifts it to a goal location.

towards the end of training, in order to optimize the environment reward. However, some examples in Figure 5 suggest that a significant repetition frequency still exists in TAAC's top-performing evaluation episodes. Note that because we take the mode of the switching policy during evaluation (Appendix C), this suggests that for those repeating steps, we have $\beta_0^* > \beta_1^*$ by the trained model. In fact, if we evaluate 100 episodes for each of the three tasks in Figure 5 and compute the action repetition percentage (repeating steps divided by total steps), the percentages are surprisingly 89%, 39%, and 49%, even though the agents are doing extremely well! A complete list of action repetition percentages for all 14 tasks can be found in Table 4. Generally, TAAC is able to adjust the repetition frequency according to tasks, resulting in a variety of frequencies across the tasks. More importantly, if we inspect the repeated actions, they are often not even close to action boundaries $\{a_{min}, a_{max}\}$, ruling out the possibility of the policy being forced to repeat due to action clipping/squashing.

We believe that there are two reasons for this surprising observation. First, with many factors such as function approximation, noise in the estimated gradient, and stochasticity of the environment dynamics, it is hardly possible for a model to reach the theoretical upper bound of an RL objective. Thus for a new action and the previous action, if their estimated Q values are very similar, then TAAC might choose to repeat with a certain chance. For example in Figure 5 (c), while the robot arm is lifting the object towards the goal in the air, it can just repeat the same lifting action for 3 steps (at $t + 5$), without losing the optimal return (up to some estimation error). Similar things happen to the bipedal walker (b) when it is jumping in the air (9 steps repeated at $t + 7$), and to the mountain car (a) when it is rushing down the hill (11 steps repeated at $t + 26$). Second, as a function approximator, the policy network $\pi_\phi$ have a limited capacity, and it is not able to represent the optimal policy at every state in a continuous space. For non-critical states that can be handled by repeating previous actions, *TAAC might learn to offload the decision making of $\pi_\phi$ onto $\beta$, and save $\pi_\phi$'s representational power for critical states*. For example, the robot arm in Figure 5 (c) invokes a fine-grained control by $\pi_\phi$ when it's grasping the object from the table, while later does not invoke $\pi_\phi$ for lifting it.

## 6 Conclusion

We have proposed TAAC, a simple but effective off-policy RL algorithm that is a middle ground between "flat" and hierarchical RL. TAAC incorporates closed-loop temporal abstraction into actor-critic by adding a second-stage policy that chooses between the previous action and a new action output by an actor. TAAC yielded strong empirical results on a variety of continuous control tasks, outperforming prior works that also model action repetition. The evaluation and visualization revealed the success factors of TAAC: persistent exploration and a compare-through Q operator for multi-step TD backup. We believe that our work has provided valuable insights into modeling temporal abstraction and action hierarchies for solving complex RL tasks in the future.

**Societal impact** This paper proposes a general algorithm to improve the efficiency of RL for continuous control. The algorithm can be applied to many robotics scenarios in the real world. How to address the potentially unsafe behaviors and risks of the deployed algorithm caused to the surroundings in real-world scenarios remains an open problem and requires much consideration.

## Acknowledgements

The authors would like to thank Jerry Bai and Le Zhao for helpful discussions on this project, and the Horizon AI platform team for infrastructure support.

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
