## A   TAAC pseudo code

---
**Algorithm 1:** Temporally abstract actor-critic
---
**Input:** $\theta$, $\phi$, $\lambda$ (learning rate), and $\tau$ (moving average rate)
**Initialize:** Randomize $\theta$ and $\phi$, $\bar{\theta} \leftarrow \theta$, $\mathcal{D} \leftarrow \emptyset$
**for** *each training iteration* **do**
    **for** *each rollout step* **do**
        $\hat{a} \sim \pi_\phi(\hat{a}|s, a^-)$              $\triangleright$ first-stage policy
        $b \sim \beta_b^*$ (Eq. 10)         $\triangleright$ second-stage policy
        $a \leftarrow a^-$ if $b = 0$ else $a \leftarrow \hat{a}$
        $s' \sim \mathcal{P}(s'|s, a)$
        $\mathcal{D} \leftarrow \mathcal{D} \bigcup \{(a^-, s, b, a, s', r(s, a, s'))\}$
        $(s, a^-) \leftarrow (s', a)$
    **end**
    **for** *each gradient step* **do**
        $\theta \leftarrow \theta - \lambda \Delta\theta$ (gradient of Eq. 3 with the compare-through $\mathcal{T}^{\pi^{\text{ta}}}$) $\triangleright$ policy evaluation
        $\phi \leftarrow \phi + \lambda \Delta\phi$ (Eq. 7)        $\triangleright$ policy improvement
        $\alpha \leftarrow \alpha - \lambda \Delta\alpha$ (gradient of Eq. 8, $\alpha = \alpha', \alpha''$)    $\triangleright$ $\alpha$ adjustment
        $\bar{\theta} \leftarrow \bar{\theta} + \tau(\theta - \bar{\theta})$        $\triangleright$ target network update
    **end**
**end**
**Output:** $\theta$ and $\phi$
---

## B   State value of the two-stage policy

Let $P(a|s, \hat{a}, a^-) \triangleq \beta_0 \delta(a - a^-) + \beta_1 \delta(a - \hat{a})$, we have

$$
\begin{aligned}
V_\theta^{\pi^{\text{ta}}}(s|a^-) &= \int_a \pi^{\text{ta}}(a|s, a^-) Q_\theta(s, a) \mathrm{d}a \\
&= \int_a \left[ \int_{\hat{a}} \pi_\phi(\hat{a}|s, a^-) P(a|s, \hat{a}, a^-) \mathrm{d}\hat{a} \right] Q_\theta(s, a) \mathrm{d}a \\
&= \int_{\hat{a}} \pi_\phi(\hat{a}|s, a^-) \left[ \int_a P(a|s, \hat{a}, a^-) Q_\theta(s, a) \mathrm{d}a \right] \mathrm{d}\hat{a} \\
&= \int_{\hat{a}} \pi_\phi(\hat{a}|s, a^-) \left[ \beta_0 Q_\theta(s, a^-) + \beta_1 Q_\theta(s, \hat{a}) \right] \mathrm{d}\hat{a} \\
&= \mathop{\mathbb{E}}_{\hat{a} \sim \pi_\phi, b \sim \beta} \left[ (1 - b) Q_\theta(s, a^-) + b Q_\theta(s, \hat{a}) \right].
\end{aligned}
$$

## C   Evaluating a policy by taking its (approximate) mode

An entropy-augmented training objective always maintains a pre-defined entropy level of the trained policy for exploration (Haarnoja et al., 2018). This results in stochastic behaviors and potentially lower scores if we measure the policy's rollout trajectories. To address this randomness issue and reflect a method's actual performance, in the experiments we evaluate a method and compute its unnormalized scores by taking the (approximate) mode of its policy distribution. Following Haarnoja et al. (2018), we use a squashed diagonal Gaussian to represent a continuous policy for every comparison method. Specifically, when sampling an action, we first sample from the unsquashed Gaussian $z \sim \mathcal{N}(\mu, \sigma^2)$, and then apply the squashing function $x = a \cdot tanh(z) + b$ to respect the action boundaries $[b - a, b + a]$. However, because of the squashing effect, it's difficult to exactly obtain the mode of this distribution. So in practice, to approximately get the mode, we first get the mode $\mu$ from the unsquashed Gaussian, and then directly apply the squashing function $\tilde{\mu} = a \cdot tanh(\mu) + b$. This $\tilde{\mu}$ is treated as the (approximate) mode of the Gaussian policy.

When evaluating TAAC, in the second stage of its two-stage policy, we also take the mode of the switching policy distribution $\beta$ as $\arg\max_{b \in \{0,1\}} \beta_b$.

# D Deriving the actor gradient

To maximize the objective in Eq. 6 with respect to $\beta$, one can parameterize $\beta$ and use stochastic gradient ascent to adjust its parameters, similar to Neunert et al. (2020). However, for every sampled $(s, a^-, \hat{a})$, there is in fact a closed-form solution for the inner expectation over $b \sim \beta$.

In general, suppose that we have $N$ values $X(i) \in \mathbb{R}, i = 0, \ldots, N-1$. We want to find a discrete distribution $P$ by the objective

$$\max_{P} \sum_{i=0}^{N-1} P(i)(X(i) - \alpha \log P(i)),$$

$$s.t. \quad \sum_{i=0}^{N-1} P(i) = 1,$$

where $\alpha > 0$. Using a Lagrangian multiplier $\lambda$, we convert it to an unconstrained optimization problem:

$$\max_{P,\lambda} \sum_{i=0}^{N-1} P(i)(X(i) - \alpha \log P(i)) + \lambda(\sum_{i=0}^{N-1} P(i) - 1).$$

Taking the derivative w.r.t. each $P(i)$ and setting it to 0, we have

$$P^*(i) = \exp\left(\frac{X(i) + \lambda - \alpha}{\alpha}\right) \propto \exp\left(\frac{X(i)}{\alpha}\right),$$

where $\lambda$ is calculated to ensure $\sum_{i=0}^{N-1} P^*(i) = 1$. Furthermore, let $Z = \sum_i \exp(\frac{X(i)}{\alpha})$ be the normalizer. The resulting maximized objective is

$$\sum_{i=0}^{N-1} P^*(i)(X(i) - \alpha(\frac{X(i)}{\alpha} - \log Z)) = \sum_{i=0}^{N-1} P^*(i)\alpha \log Z = \alpha \log Z. \tag{9}$$

To derive $\beta^*$ for Eq. 6 given any $(s, a^-) \sim \mathcal{D}, \hat{a} \sim \pi_\phi$, we set $X(0) = Q_\theta(s, a^-)$ and $X(1) = Q_\theta(s, \hat{a})$. Then $\beta^*$ can be found as below:

$$\beta_b^* \propto \exp\left(\frac{(1-b)Q_\theta(s, a^-) + bQ_\theta(s, \hat{a})}{\alpha}\right). \tag{10}$$

Since this is a global maximum solution given any sampled $(s, a^-, \hat{a})$, $\beta^*$ is guaranteed to be no worse than any parameterized policy. Putting $\beta^*$ back into Eq. 6, we are able to simplify $V_\theta^{\pi^{\text{ta}}}$ (referring to Eq. 9) as

$$\mathbb{E}_{\hat{a} \sim \pi_\phi} \alpha \left[\log \sum_{b=0}^{1} \exp\left(\frac{(1-b)Q_\theta(s, a^-) + bQ_\theta(s, \hat{a})}{\alpha}\right) - \log \pi_\phi(\hat{a}|s, a^-)\right]. \tag{11}$$

Then we apply the re-parameterization trick $\hat{a} = f_\phi(\epsilon, s, a^-), \epsilon \sim \mathcal{N}(0, I)$. Approximating the gradient w.r.t. $\phi$ with a single sample of $\epsilon$, we get Eq. 7.

# E Multi-step actor gradient

Let us first consider on-policy training as a simplified setting when computing the actor gradient for our two-stage policy $\pi^{\text{ta}}$. Suppose the unroll length is $M$, which means each time we unroll the current policy $\pi^{\text{ta}}$ for $M$ steps, do a gradient update with the collected data, and unroll with the updated policy for the next $M$ steps, and so on. In this case, the rollout computational graph is illustrated in Figure 6. Let $\beta_b^*(t + m) = \beta^*(b|s_{t+m}, \hat{a}_{t+m}, a_{t+m}^-)$ be the optimal $\beta$ policy at step $t + m$ and define $w_{t+m} = \beta_1^*(t) \prod_{m'=1}^{m} \beta_0^*(t + m')$ to be the probability of $V_\theta^{\pi^{\text{ta}}}(s_{t+m}|a_{t+m}^-)$ adopting $\hat{a}_t$ via action repetition. The overall V value over the $M$ steps to be maximized is

$$\mathcal{V} \triangleq \sum_{m=0}^{M-1} V_\theta^{\pi^{\text{ta}}}(s_{t+m}|a_{t+m}^-).$$

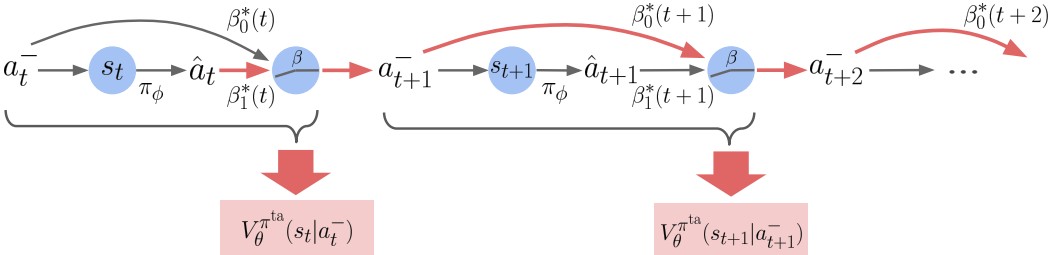

Figure 6: Rollout computational graph of $\pi^{\text{ta}}$ starting from step $t$. Rectangles denote negative losses to maximize. The red edges denote the gradient paths (reversed direction) for $\hat{a}_t$ when maximizing the sum of all V values: $\sum_{m=0}^{M-1} V_\theta^{\pi^{\text{ta}}}(s_{t+m}|a_{t+m}^-)$.

Ignoring the entropy term, its gradient w.r.t. $\hat{a}^t$ is

$$\frac{\partial \mathcal{V}}{\partial \hat{a}^t} = \sum_{m=0}^{M-1} w_{t+m} \frac{\partial Q_\theta(s_{t+m}, \hat{a}_t)}{\partial \hat{a}_t}, \tag{12}$$

where the weights $w_{t+m}$ correspond to different red partial paths staring from $\hat{a}^t$ and ending at different negative losses in Figure 6.

The above computational graph assumes that when unrolling the current $\pi^{\text{ta}}$, we are able to interact with the environment to obtain states $\{s_{t+1}, s_{t+2}, \ldots\}$. This is true for on-policy training but not for off-policy training. In the latter case, if we directly use the sampled sequence $\{s_t, s_{t+1}, s_{t+2}, \ldots\}$ from the replay buffer in Eq. 12, the resulting gradient will suffer from off-policyness. Thus in practice, we truncate the gradient to the first step, $\frac{\partial \mathcal{V}}{\partial \hat{a}_t} \approx w_t \frac{\partial Q_\theta(s_t, \hat{a}_t)}{\partial \hat{a}_t}$. We believe that this truncation is a simple but good approximation, for the two reasons:

a) Because Eq. 12 is defined for a sampled state-action trajectory, $\frac{\partial Q_\theta(s_{t+m}, \hat{a}_t)}{\partial \hat{a}_t}$ has a much higher sample variance as $m$ increases.

b) The weights $w_{t+m}$ decrease exponentially so the influence of $\hat{a}_t$ on future $\frac{\partial Q_\theta}{\partial \hat{a}_t}$ quickly decays.

Empirically, the truncated one-step gradient yields good results in our experiments.

## F    Different temperatures for $\beta$ and $\pi_\phi$

We use two different temperatures $\alpha'$ and $\alpha''$ for weighting the entropy of $\beta$ and $\pi_\phi$ respectively, to have a finer control of their entropy terms. Accordingly, the objective in Eq. 6 changes to

$$\mathbb{E}_{(s,a^-)\sim\mathcal{D},\hat{a}\sim\pi_\phi,b\sim\beta} \left[ (1-b)Q_\theta(s,a^-) + bQ_\theta(s,\hat{a}) - \alpha' \log \beta_b - \alpha'' \log \pi_\phi(\hat{a}|s,a^-) \right].$$

Revisiting Section 4.3, several key formulas are updated to reflect this change. Eq. 10 is updated to

$$\beta_b^* \propto \exp\left( \frac{(1-b)Q_\theta(s,a^-) + bQ_\theta(s,\hat{a})}{\alpha'} \right). \tag{13}$$

Eq. 7 is updated to

$$\Delta\phi \triangleq \left( \beta_1^* \frac{\partial Q_\theta(s,\hat{a})}{\partial \hat{a}} - \alpha'' \frac{\partial \log \pi_\phi(\hat{a}|s,a^-)}{\partial \hat{a}} \right) \frac{\partial f_\phi}{\partial \phi} - \alpha'' \frac{\partial \log \pi_\phi(\hat{a}|s,a^-)}{\partial \phi}.$$

## G    SAC-Krep

Following Delalleau et al. (2020) we extend the original SAC algorithm to support a hybrid of discrete and continuous actions [2], to implement the baseline SAC-Krep (Sharma et al., 2017; Biedenkapp

---

[2]An implementation of SAC with hybrid actions is available at https://github.com/HorizonRobotics/alf/blob/pytorch/alf/algorithms/sac_algorithm.py.

et al., 2021) in Section 5.2. We denote the discrete and continuous actions by $b$ $(1 \leq b \leq B)$ and $a$, respectively. Let the joint policy be $\pi(a, b|s) = \pi_\phi(a|s)\pi(b|s, a)$, namely, the joint policy is decomposed in a way that it outputs a continuous action, followed by a discrete action conditioned on that continuous action. Let $Q_\theta(s, a, b)$ be the parameterized expected return of taking action $(a, b)$ at state $s$. Then the entropy-augmented state value is computed as

$$V_\theta^\pi(s) = \mathop{\mathbb{E}}_{(a,b) \sim \pi} \Big[ Q_\theta(s, a, b) - \alpha'' \log \pi_\phi(a|s) - \alpha' \log \pi(b|s, a) \Big].$$

Similar to Section 4.3, we can derive an optimal closed-form $\pi^*(b|s, a)$ given any $(s, a)$, and then optimize the continuous policy $\pi_\phi(a|s)$ similarly to Eq. 7.

For policy evaluation, in the case of SAC-Krep, $b$ represents how many steps $a$ will be executed without being interrupted. Thus the objective of learning $Q_\theta$ is

$$\min_\theta \mathop{\mathbb{E}}_{(s_t, a_t, b_t, s_{t+b_t}) \sim \mathcal{D}} \left[ Q_\theta(s_t, a_t, b_t) - \mathcal{B}^\pi Q_{\bar\theta}(s_t, a_t, b_t) \right]^2,$$

$$\text{with } \mathcal{B}^\pi Q_{\bar\theta}(s_t, a_t, b_t) = \sum_{t'=t}^{t+b_t-1} \gamma^{t'-t} r(s_{t'}, a_{t'}, s_{t'+1}) + \gamma^{b_t} V_{\bar\theta}^\pi(s_{t+b_t}),$$

Namely, the Q value is bootstrapped by $b$ steps. We instantiate the Q network by having the continuous action $a$ as an input in addition to $s$, and let the network output $B$ heads, each representing $Q(s, a, b)$.

# H SAC-Hybrid

Following H-MPO (Neunert et al., 2020) we define a factored policy of a newly sampled continuous action $\hat{a}$ and a binary switching action $b$:

$$\pi((\hat{a}, b)|s, a^-) = \pi_{\phi_a}(\hat{a}|s, a^-)\pi_{\phi_b}(b|s, a^-),$$

where the observation consists of state $s$ and previous action $a^-$. Note that a big difference between this formulation with either TAAC or SAC-Krep (Appendix G) is that $\hat{a}$ and $b$ are independent. That is, the decision of "repeat-or-act" is made in parallel with the new action. The entropy-augmented state value is computed as

$$V_\theta^\pi((s, a^-)) = \mathop{\mathbb{E}}_{\hat{a} \sim \pi_{\phi_a}, b \sim \pi_{\phi_b}} \Big[ Q_\theta((s, a^-), (\hat{a}, b)) - \alpha'' \log \pi_{\phi_a}(\hat{a}|s, a^-) - \alpha' \log \pi_{\phi_b}(b|s, a^-) \Big],$$

and $\phi_a$ and $\phi_b$ can be optimized by gradient ascent. Finally, the Bellman operator for policy evaluation is

$$\mathcal{B}^\pi Q_{\bar\theta}((s, a^-), (\hat{a}, b)) = r(s, a, s') + \gamma V_{\bar\theta}^\pi((s', a)),$$

where $a = (1-b)a^- + b\hat{a}$ is the action output to the environment. Similar to SAC-Krep, to instantiate the Q network, we use $(s, a^-, \hat{a})$ as the inputs and let the network output two heads for $b = 0$ and $b = 1$. Compared to TAAC's Q formulation (Eq. 3), clearly SAC-Hybrid's Q has to handle more input/output mappings for the same transition dynamics, which makes the policy evaluation less efficient.

# I Task details

All 14 tasks are wrapped by the OpenAI Gym (Brockman et al., 2016) interface. All of them, except *Town01*, are very standard and follow their original definitions. The environment of *Town01* is customized by us with various map options using a base map called "Town01" provided by the CARLA simulator (Dosovitskiy et al., 2017), which we will describe in detail later. We always scale the action space of every task to $[-1, 1]^A$, where $A$ is the action dimensionality defined by the task environment. The observation space of each task is unchanged, except for *Town01*. Note that we use MuJoCo 2.0 (Todorov et al., 2012) for simulating **Locomotion** and **Manipulation** [3]. A summary of the tasks is in Table 5.

---

[3]A different version of MuJoCo may result in different observations, rewards, and incomparable environments; see https://github.com/openai/gym/issues/1541.

| Category | Task | Gym environment name | Observation space | Action space | Reward normalization |
|---|---|---|---|---|---|
| **SimpleControl** | *MountainCarContinuous* | `MountainCarContinuous-v0` | $\mathbb{R}^2$ | $[-1,1]^1$ | $[-5,5]$ |
| | *LunarLanderContinuous* | `LunarLanderContinuous-v2` | $\mathbb{R}^8$ | $[-1,1]^2$ | |
| | *InvertedDoublePendulum* | `InvertedDoublePendulum-v2` | $\mathbb{R}^{11}$ | $[-1,1]^1$ | |
| **Locomotion** | *Hopper* | `Hopper-v2` | $\mathbb{R}^{11}$ | $[-1,1]^3$ | $\times$ |
| | *Ant* | `Ant-v2` | $\mathbb{R}^{111}$ | $[-1,1]^8$ | |
| | *Walker2d* | `Walker2d-v2` | $\mathbb{R}^{17}$ | $[-1,1]^6$ | |
| | *HalfCheetah* | `HalfCheetah-v2` | | | |
| **Terrain** | *BipedalWalker* | `BipedalWalker-v2` | $\mathbb{R}^{24}$ | $[-1,1]^4$ | $[-1,1]$ |
| | *BipedalWalkerHardcore* | `BipedalWalkerHardcore-v2` | | | |
| **Manipulation** | *FetchReach* | `FetchReach-v1` | $\mathbb{R}^{13}$ | $[-1,1]^4$ | $[-1,1]$ |
| | *FetchPush* | `FetchPush-v1` | $\mathbb{R}^{28}$ | | |
| | *FetchSlide* | `FetchSlide-v1` | | | |
| | *FetchPickAndPlace* | `FetchPickAndPlace-v1` | | | |
| **Driving** | *Town01* | `Town01` | "camera": $\mathbb{R}^{128\times64\times3}$, "radar": $\mathbb{R}^{200\times4}$, "collision": $\mathbb{R}^{4\times3}$, "IMU": $\mathbb{R}^7$, "goal": $\mathbb{R}^3$, "velocity": $\mathbb{R}^3$, "navigation": $\mathbb{R}^{8\times3}$ "prev action": $[-1,1]^4$ | $[-1,1]^4$ | $[-5,5]$ |

Table 5: The 14 tasks with their environment details. Note that reward clipping is performed after reward normalization (if applied). Except *Town01*, the input observation is a flattened vector.

## I.1 Reward normalization

We normalize each task's reward using a normalizer that maintains adaptive exponential moving averages of the reward and its second moment. Specifically, let $\xi$ be a pre-defined update speed ($\xi = 8$ across all experiments), and $L$ be the total number of times the normalizer statistics has been updated so far, then for the incoming reward $r$, the mean $m_1$ and second moment $m_2$ are updated as

$$
\begin{aligned}
\eta_L &= \frac{\xi}{L+\xi}, \\
m_1 &\leftarrow (1 - \eta_L)m_1 + \eta_L r, \\
m_2 &\leftarrow (1 - \eta_L)m_2 + \eta_L r^2, \\
L &\leftarrow L + 1,
\end{aligned}
$$

with $L = m_1 = m_2 = 0$ as the initialized values. Basically, the moving average rate $\eta_L$ decreases according to $\frac{1}{L}$. With this averaging strategy, one can show that by step $L$, the weight for the reward encountered at step $l \leq L$ is roughly in proportional to $(\frac{l}{L})^{(\xi-1)}$. Intuitively, as $L$ increases, the effective averaging window expands because the averaging weights are computed by the changing ratio $\frac{l}{L}$. Finally, given any reward $r'$, it is normalized as

$$
\min(\max(\frac{r' - m_1}{\sqrt{m_2 - m_1^2}}, -c), c),
$$

where $c > 0$ is a constant set to either 1 or 5, according to which value produces better performance for SAC on a task. We find that the suite of **Locomotion** tasks is extremely sensitive to reward definition, and thus do not apply reward normalization to it. The normalizer statistics is updated only when rewards are sampled from the replay buffer. Note that for a task, the same reward normalization (if applied) is used by all 8 evaluated methods with no discrimination.

## I.2 Town01

Our *Town01* task is based on the "Town01" map (Dosovitskiy et al., 2017) that consists of 12 T-junctions (Figure 7). The map size is roughly $400 \times 400 \text{ m}^2$. At the beginning of each episode, the vehicle is first randomly spawned at a lane location. Then a random waypoint is selected on the map and is set as the destination for the vehicle. The maximal episode length (time limit) is computed as

$$
N_{\text{frames}} = \frac{L_{\text{route}}}{S_{\text{min}} \cdot \Delta t}
$$

where $L_{\text{route}}$ is the shortest route length calculated by the simulator, $S_{\text{min}}$ is the average minimal speed expected for a meaningful driving, and $\Delta t$ is the simulation step time. We set $S_{\text{min}} = 5\text{m/s}$ and $\Delta t = 0.1\text{s}$ through the experiment. An episode can terminate early if the vehicle reaches the

destination, or gets stuck at collision for over a certain amount of time. We customize the map to include 20 other vehicles and 20 pedestrians that are programmed by the simulator's built-in AI to act in the scenario. We use the default weather type and set the day length to 1000 seconds.

The action space of the vehicle is 4 dimensional: ("throttle", "steer", "brake", "reverse"). We customize the observation space to include 8 different multi-modal inputs:

1. "camera": a monocular RGB image ($128 \times 64 \times 3$) that shows the road condition in front of the vehicle;
2. "radar": an array of 200 radar points, where each point is represented by a 4D vector;
3. "collision": an array of 4 collisions, where each collision is represented by a 3D vector;
4. "IMU": a 7D IMU measurement vector of the vehicle's status;
5. "goal": a 3D vector indicating the destination location;
6. "velocity": the velocity of the vehicle relative to its own coordinate system;
7. "navigation": an array of 8 future waypoints on the current navigation route, each waypoint is a 3D vector in the coordinate system of the vehicle;
8. "prev action": the action taken by the vehicle at the previous time step.

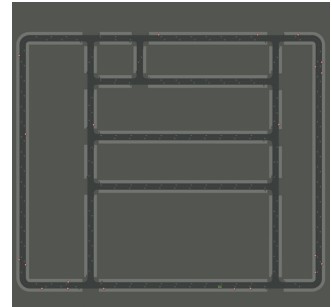

Figure 7: The layout of the map "Town01" (picture from https://carla.readthedocs.io). The actual map is filled with other objects such as buildings, trees, pedestrians, and traffic lights to make it a realistic scene of a small town.

Since the observation space of *Town01* is huge, we apply normalization to all input sensors for more efficient training. We normalize each input sensor vector in a similar way of reward normalization. After normalization, the vector is element-wisely clipped to $[-5, 5]$.

To train the vehicle, we define the task reward by 4 major components:

1. "distance": a shaped reward that measures how much closer the vehicle is to the next navigation waypoint after one time step;
2. "collision": if a collision is detected, then the vehicle gets a reward of $-\min(20, 0.5 \cdot \max(0, \bar{R}))$, where $\bar{R}$ is the accumulated episode reward so far;
3. "red light": if a red light violation is detected, then the vehicle gets a reward of $-\min(20, 0.3 \cdot \max(0, \bar{R}))$, where $\bar{R}$ is the accumulated episode reward so far;
4. "goal": the vehicle gets a reward of $100$ for reaching the destination.

The overall reward at a time step is computed as the sum of the above 4 rewards. This reward definition ensures that SAC obtains reasonable performance in this task.

## I.3 Network structure

The model architecture of all compared methods is identical: each trains an actor network and a critic network [4]. Following Fujimoto et al. (2018), the critic network utilizes two replicas to reduce positive bias in the Q function. We make sure that the actor and critic network always have the same structure (but with different weights) except for the final output layer.

Below we review the network structure designed for each task category, shared between the actor and critic network, and shared among all 8 evaluated methods. No additional network or layer is owned exclusively by any method.

− **SimpleControl**: two hidden layers, each of size 256.
− **Locomotion**: two hidden layers, each of size 256.
− **Terrain**: two hidden layers, each of size 256.
− **Manipulation**: three hidden layers, each of size 256.

---

[4]SAC-Krep actually has an extra discrete Q network that models the values of repeating $1, 2, \ldots, N$ steps. It has the same structure with the critic network for the continuous action, but with multiple output heads.

| Hyperparameter | SimpleControl | Terrain | Driving | Manipulation (Plappert et al., 2018) | Locomotion (Haarnoja et al., 2018) |
|---|---|---|---|---|---|
| Learning rate | $10^{-4}$ | $5 \times 10^{-4}$ | | $10^{-3}$ | $3 \times 10^{-4}$ |
| Reward discount | 0.99 | | | 0.98 | |
| Number of parallel actors for rollout | 1 | 32 | 4 | 38 | |
| Replay buffer size per actor | $10^5$ | | | $2 \times 10^4$ | $10^6$ |
| Mini-batch size | 256 | 4096 | 64 | 4864 | |
| Entropy target $\delta$ (Eq. 14) | 0.1 | | | 0.2 | 0.184 |
| Target Q smoothing coefficient | $5 \times 10^{-3}$ | | | $5 \times 10^{-2}$ | |
| Target Q update interval | 1 | | | 40 | |
| Training interval (env frames) per actor | 1 | 5 | 10 | 50/40 | |
| Total environment frames for rollout | $10^5$ | $5 \times 10^6$ | $10^7$ | $10^7$ | $10^6$ |

Table 6: The hyperparameter values of SAC on 5 task categories. The two shaded columns **Manipulation** and **Locomotion** use exactly the same hyperparameter values from the original papers, and we list them for completeness. An empty cell in the table means using the same hyperparameter value as the corresponding one of **SimpleControl**. The training interval of **Manipulation** ($50/40$) means updating models 40 times in a row for every 50 environment steps (per actor), which also follows the convention set by Plappert et al. (2018).

- **Driving**: We use an encoder to combine multi-modal sensor inputs. The encoder uses a mini ResNet (He et al., 2016) of 6 bottleneck blocks (without BatchNorm) to encode an RGB image into a latent embedding of size 256, where each bottleneck block has a kernel size of 3, filters of $(64, 32, 64)$, and a stride of 2 (odd block) or 1 (even block). The encoder then flattens any other input and projects it to a latent embedding of size 256. All the latent embeddings are averaged and input to an FC layer of size 256 to yield a single encoded vector that summarizes the input sensors. Finally, the actor/critic network is created with one hidden layer of size 256, with this common encoded vector as input. We detach the gradient when inputting the encoded vector to the actor network, and only allow the critic network to learn it.

We use ReLU for all hidden activations.

## J Experiment details

### J.1 Entropy target calculation

When computing an entropy target, instead of directly specifying a floating number which is usually unintuitive, we calculate it by an alternative parameter $\delta$. If the action space is continuous, then suppose that it has $K$ dimensions, and every dimension is bounded by $[m, M]$. We assume the entropy target to be the entropy of a continuous distribution whose probability uniformly concentrates on a slice of the support $\delta(M - m)$ with $P = \frac{1}{\delta(M-m)}$. Thus the entropy target is calculated as

$$-K \int_m^M P(a) \log P(a) \mathrm{d}a = -K \log \frac{1}{\delta(M - m)} = K \left[ \log \delta + \log(M - m) \right]. \tag{14}$$

For example, by this definition, an entropy target of -1 per dimension used by Haarnoja et al. (2018) is equivalent to setting $\delta = 0.184$ here with $M = 1$ and $m = -1$. If the action space is discrete with $K > 1$ entries, we assume the entropy target to be the entropy of a discrete distribution that has one entry of probability $1 - \delta$, with $\delta$ uniformly distributed over the other $K - 1$ entries. Thus the entropy target is calculated as

$$-\delta \log \frac{\delta}{K - 1} - (1 - \delta) \log(1 - \delta). \tag{15}$$

We find setting $\delta$ instead of the direct entropy target is always more intuitive in practice.

### J.2 Hyperparameters

We use Adam (Kingma and Ba, 2015) with $\beta_1 = 0.9$, $\beta_2 = 0.999$, and $\epsilon = 10^{-7}$ to train each method. Below we first describe the hyperparameter values of the vanilla SAC baseline. These values are selected by referring to either previously published ones or our typical options for SAC runs. For **Locomotion** and **Manipulation**, we directly adopt the hyperparameter values from the original papers. Among the remaining 3 task categories (**SimpleControl**, **Terrain**, and **Driving**), several

hyperparameter values vary due to task differences (Table 6). This variance also serves to test if our comparison result generalizes under different training settings.

The hyperparameter values of the other 7 evaluated methods are the same as shown in Table 6, but with extra hyperparameters if required by a method. Note that for open-loop action repetition methods like SAC-Nrep and SAC-Krep, the counting of environment frames includes frames generated by repeated actions. For the repeating hyperparameter $N$ in Section 5.2, we set it to 3 for **SimpleControl**, **Locomotion** and **Manipulation**, and to 5 for **Terrain** and **Driving**. For SAC-EZ, we set $\mu$ of the zeta distribution to 2 following Dabney et al. (2021), and linearly decay $\epsilon$ from 1 to 0.01 over the course of first $\frac{1}{10}$ of the training. $\epsilon$ is then kept to be 0.01 till the end of training. Finally, for SAC-Hybrid, TAAC-1td, TAAC-Ntd, and TAAC, the discrete action requires its own entropy target (Appendix F) computed by Eq. 15. We set $\delta$ in that equation to 0.05 in all task categories. In **Locomotion**, we clip the advantage $Q_\theta(s, \hat{a}) - Q_\theta(s, a^-)$ to $[0, +\infty)$ when computing $\beta^*$ for TAAC-1td, TAAC-Ntd, and TAAC. This clipping biases the agent towards sampling new actions.

### J.3 Computational resources

The required computational resources for doing all our experiments are moderate. We define a job group as a (`method`, `task_category`) pair, *e.g.*, (TAAC, **Locomotion**). We used three RTX 2080Ti GPUs at the same time for training the jobs within one group simultaneously. We evenly distributed a job group (each job in the group represents a (`task`, `random_seed`) combination, *e.g.*, (*Ant*, *seed0*)) across the three GPUs. Finally, all job groups were launched on a cluster. Among the groups, (TAAC, **Driving**) took the longest training time which was roughly 36 hours, while (SAC, **SimpleControl**) took the shortest time which was about 2 hours. The rest of job groups mostly finished within 12 hours each.

## K More experimental results

For the 8 comparison methods in Section 5.2, we list their n-score curves of all 5 task categories in Figure 8, and their unnormalized score curves of all 14 tasks in Figure 9. The score curves are smoothed using exponential moving average to reduce noises.

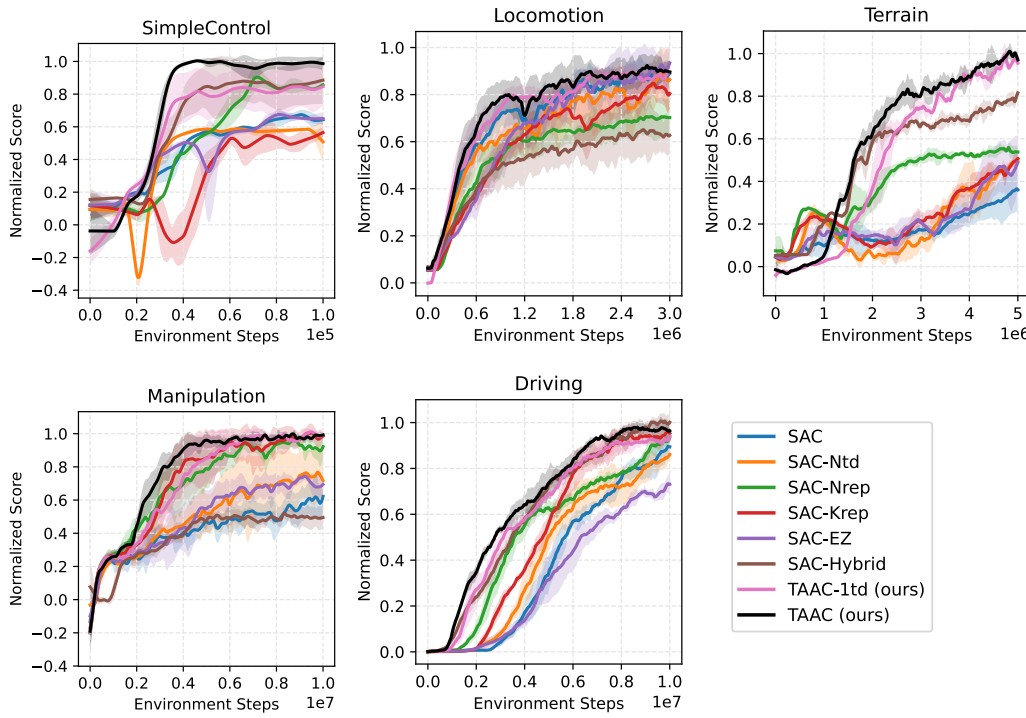

Figure 8: The n-score curves of the 5 task categories. Each curve is a mean of a method's n-score curves on the tasks within a task category, where the method is run with 3 random seeds for each task.

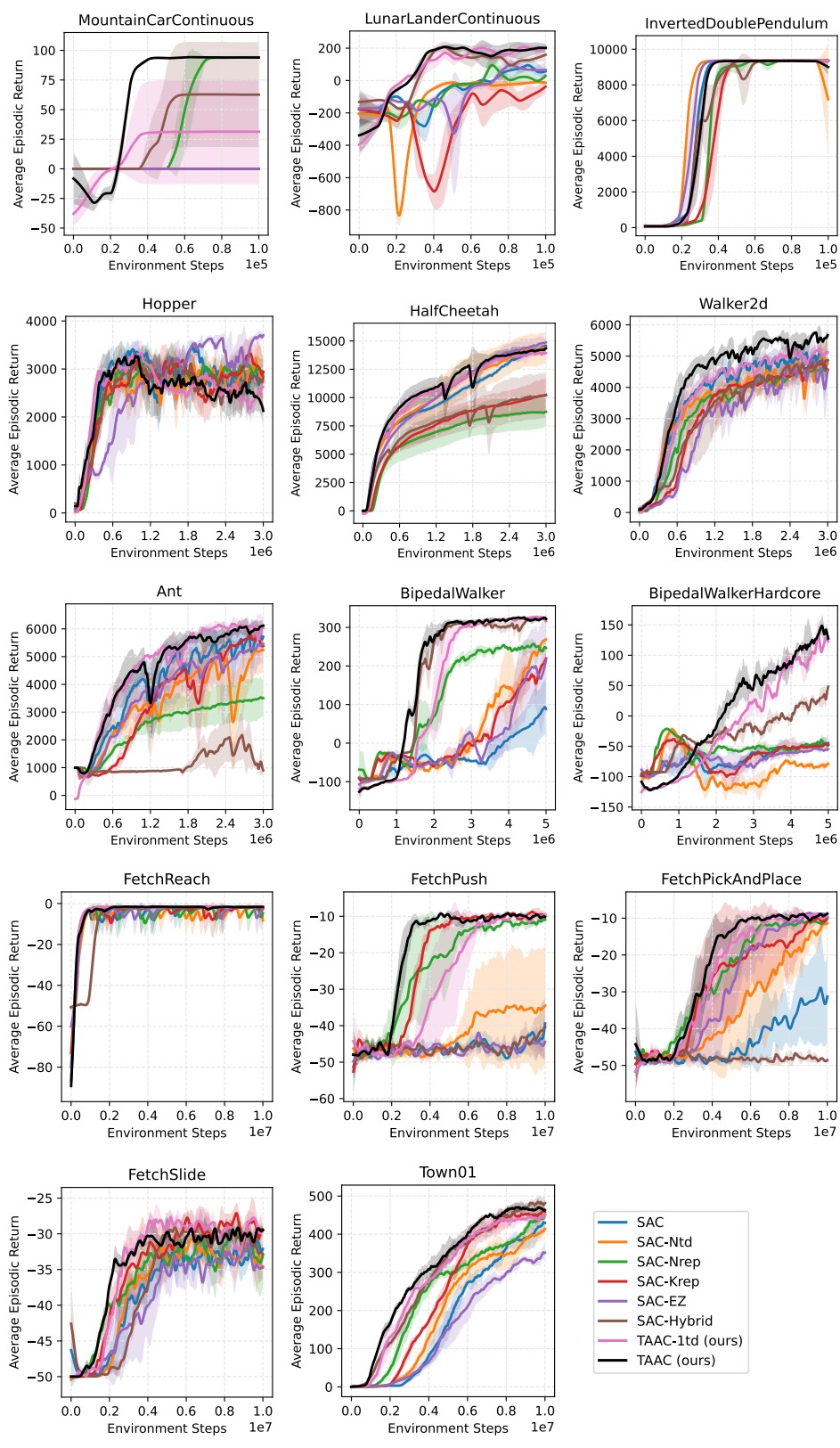

Figure 9: The unnormalized reward curves of the 14 tasks. Each curve is averaged over 3 random seeds, and the shaded area around it represents the standard deviation.

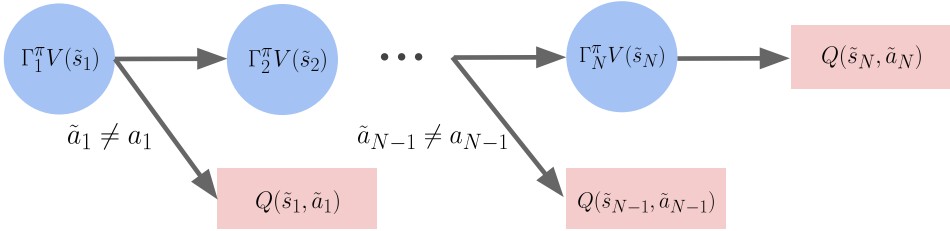

Figure 10: The stochastic binary tree defined by $\mathcal{T}^\pi$. Circles are inner nodes and rectangles are leaves. Whenever the sampled action $\tilde{a}_n$ is not equal to the rollout action $a_n$, a tree path terminates.

## L   Proof of multi-step policy evaluation convergence

We now prove that the compare-through Q operator $\mathcal{T}^\pi$ in Section 4.2 is unbiased, namely, $Q^\pi$ is a unique fixed point of $\mathcal{T}^\pi$ (policy evaluation convergence). Assuming a tabular setting, value functions and policies are no longer parameterized and can be enumerated over all states and actions.

### L.1   Definition

We first present a formal definition of $\mathcal{T}^\pi$. Suppose that we consider $N$-step ($N \geq 1$) TD learning. Each time we sample a historical trajectory $(s_0, a_0, s_1, r_{s_0,a_0,s_1}, \ldots, s_N, r_{s_{N-1},a_{N-1},s_N})$ of $N + 1$ steps from the replay buffer to update $Q(s_0, a_0)$. For convenience, we define a sequence of auxiliary operators $\Gamma_n^\pi$ for the $V$ values backup recursively as

$$\Gamma_N^\pi V(\tilde{s}_N) = \mathop{\mathbb{E}}_{\pi(\tilde{a}_N|\tilde{s}_N)} Q(\tilde{s}_N, \tilde{a}_N),$$

$$\Gamma_n^\pi V(\tilde{s}_n) = \mathop{\mathbb{E}}_{\pi(\tilde{a}_n|\tilde{s}_n)} \left[ \mathbb{1}_{\tilde{a}_n \neq a_n} \underbrace{Q(\tilde{s}_n, \tilde{a}_n)}_{\text{"stop"}} + \mathbb{1}_{\tilde{a}_n = a_n} \underbrace{\mathop{\mathbb{E}}_{\mathcal{P}(\tilde{s}_{n+1}|\tilde{s}_n,\tilde{a}_n)} \left[ r_{\tilde{s}_n,\tilde{a}_n,\tilde{s}_{n+1}} + \gamma \Gamma_{n+1}^\pi V(\tilde{s}_{n+1}) \right]}_{\text{"expand"}} \right],$$

for $1 \leq n \leq N - 1,$

$$(16)$$

and based on which we define

$$\mathcal{T}^\pi Q(s_0, a_0) = \mathop{\mathbb{E}}_{\mathcal{P}(\tilde{s}_1|s_0,a_0)} \left[ r_{s_0,a_0,\tilde{s}_1} + \gamma \Gamma_1^\pi V(\tilde{s}_1) \right] \qquad (17)$$

as the final operator to update $Q(s_0, a_0)$. Intuitively, the above recursive transform defines a stochastic binary tree, where $\Gamma_n^\pi V(\cdot)$ are inner nodes and $Q(\cdot)$ are leaves (Figure 10). The branching of an inner node (except the last $\Gamma_N^\pi V(\tilde{s}_N)$) depends on the indicator function $\mathbb{1}_{\tilde{a}_n \neq a_n}$. From the root $\Gamma_1^\pi V(\tilde{s}_1)$ to a leaf, the maximum path length is $N + 1$ (when all $\tilde{a}_n = a_n$) and the minimum path length is 2 (when $\tilde{a}_1 \neq a_1$).

To actually estimate $\mathcal{T}^\pi Q(s_0, a_0)$ during off-policy training without access to the environment for $\mathcal{P}$ and $r$, we use the technique introduced in Section 4.2 to sample a path from the root to a leaf on the binary tree, by re-using the historical trajectory as much as possible. Specifically, we first set $\tilde{s}_1 = s_1$ and $r_{s_0,a_0,\tilde{s}_1} = r_{s_0,a_0,s_1}$ as in the typical 1-step TD learning setting. Starting from $n = 1$, we sample $\tilde{a}_n \sim \pi(\cdot|s_n)$ and compare $\tilde{a}_n$ with $a_n$. If they are equal, we continue to set $\tilde{s}_{n+1} = s_{n+1}$ and $r_{\tilde{s}_n,\tilde{a}_n,\tilde{s}_{n+1}} = r_{s_n,a_n,s_{n+1}}$. We repeat this process until $\tilde{a}_n \neq a_n$. In a word,

$$\mathcal{T}^\pi Q(s_0, a_0) \approx r_{s_0,a_0,s_1} + \gamma r_{s_1,a_1,s_2} + \ldots + \gamma^n Q(s_n, \tilde{a}_n), \ n = \min\left(\{n|\tilde{a}_n \neq a_n\} \cup \{N\}\right).$$

Usually for a continuous policy $\pi$, $\mathbb{1}_{\tilde{a}_n \neq a_n}$ is 1 with a probability of 1 because two sampled actions are always unequal. So $\mathcal{T}^\pi$ will stop expanding at $s_1$ and it seems no more than just a normal Bellman operator for 1-step TD learning. However, if $\pi$ is specially structured and has a way of generating two identical actions in a continuous space, then it has the privilege of entering deeper tree branches for multi-step TD learning. For example, TAAC is indeed able to generate $\tilde{a}_n$ identical to the rollout action $a_n$ if $\tilde{b}_k = b_k = 0$, for all $1 \leq k \leq n$. In this case, $\tilde{a}_n = a_n = a_0$.

Here we note that the above point estimate of $\mathcal{T}^\pi$ can also be written as $\mathcal{T}^\pi Q(s_0, a_0) \approx Q(s_0, a_0) + \Delta Q(s_0, a_0)$, where

$$\Delta Q(s_0, a_0) = \sum_{n=0}^{N-1} \gamma^n \left( \prod_{i=0}^{n} \mathbb{1}_{a_i = \tilde{a}_i} \right) \left[ r_{s_n, a_n, s_{n+1}} + \gamma Q(s_{n+1}, \tilde{a}_{n+1}) - Q(s_n, \tilde{a}_n) \right].$$

Thus it shares a very similar form with Retrace (Munos et al., 2016), except the traces are now binary values defined by action comparison.

## L.2 Convergence proof

Given any historical trajectory $\tau = (s_0, a_0, s_1, r_{s_0, a_0, s_1}, \ldots, s_N, r_{s_{N-1}, a_{N-1}, s_N})$ from an arbitrary behavior policy, we first verify that $Q^\pi$ is a fixed point of $\mathcal{T}^\pi$. When $Q = Q^\pi$ in Eq. 16, we have $\Gamma_N^\pi V(\tilde{s}_n) = \mathbb{E}_{\pi(\tilde{a}_N | \tilde{s}_n)} Q^\pi(\tilde{s}_n, \tilde{a}_N) = V^\pi(\tilde{s}_n)$. Now assuming $\Gamma_{n+1}^\pi V = V^\pi$, we have

$$
\begin{aligned}
\Gamma_n^\pi V(\tilde{s}_n) &= \mathop{\mathbb{E}}_{\pi(\tilde{a}_n | \tilde{s}_n)} \left[ \mathbb{1}_{\tilde{a}_n \neq a_n} \cdot Q^\pi(\tilde{s}_n, \tilde{a}_n) + \mathbb{1}_{\tilde{a}_n = a_n} \mathop{\mathbb{E}}_{\mathcal{P}(\tilde{s}_{n+1} | \tilde{s}_n, \tilde{a}_n)} \left[ r_{\tilde{s}_n, \tilde{a}_n, \tilde{s}_{n+1}} + \gamma V^\pi(\tilde{s}_{n+1}) \right] \right] \\
&= \mathop{\mathbb{E}}_{\pi(\tilde{a}_n | \tilde{s}_n)} \left[ \mathbb{1}_{\tilde{a}_n \neq a_n} Q^\pi(\tilde{s}_n, \tilde{a}_n) + \mathbb{1}_{\tilde{a}_n = a_n} Q^\pi(\tilde{s}_n, \tilde{a}_n) \right] \\
&= \mathop{\mathbb{E}}_{\pi(\tilde{a}_n | \tilde{s}_n)} Q^\pi(\tilde{s}_n, \tilde{a}_n) \\
&= V^\pi(\tilde{s}_n).
\end{aligned}
$$

Thus finally we have $\mathcal{T}^\pi Q^\pi(s_0, a_0) = \mathbb{E}_{\mathcal{P}(\tilde{s}_1 | s_0, a_0)}[r_{s_0, a_0, \tilde{s}_1} + \gamma V^\pi(\tilde{s}_1)] = Q^\pi(s_0, a_0)$ for any $(s_0, a_0)$.

To prove that $Q^\pi$ is the unique fixed point of $\mathcal{T}^\pi$, we verify that $\mathcal{T}^\pi$ is a contraction mapping on the infinity norm space of $Q$. Suppose we have two different Q instantiations $Q$ and $Q'$, we would like prove that after applying $\mathcal{T}^\pi$ to them, $\|Q - Q'\|_\infty$ becomes strictly smaller than before. Let $\Delta = \|Q - Q'\|_\infty$ be the current infinity norm, i.e., $\Delta = \max_{s,a} |Q(s,a) - Q'(s,a)|$. Then we have

$$
\begin{aligned}
\|\Gamma_N^\pi V - \Gamma_N^\pi V'\|_\infty &= \max_s |\Gamma_N^\pi V(s) - \Gamma_N^\pi V'(s)| \\
&= \max_s \left| \mathop{\mathbb{E}}_{\pi(\cdot|s)} (Q(s, \cdot) - Q'(s, \cdot)) \right| \\
&\leq \max_s \mathop{\mathbb{E}}_{\pi(\cdot|s)} |Q(s, \cdot) - Q'(s, \cdot)| \\
&\leq \max_s \mathop{\mathbb{E}}_{\pi(\cdot|s)} \Delta \\
&= \Delta,
\end{aligned}
$$

and for $1 \leq n \leq N - 1$ recursively

$$
\begin{aligned}
\|\Gamma_n^\pi V - \Gamma_n^\pi V'\|_\infty &= \max_s |\Gamma_n^\pi V(s) - \Gamma_n^\pi V'(s)| \\
&= \max_s \left| \mathop{\mathbb{E}}_{\pi(a|s)} \left[ \mathbb{1}_{a \neq a_n}(Q(s,a) - Q'(s,a)) + \mathbb{1}_{a = a_n} \gamma \mathop{\mathbb{E}}_{\mathcal{P}(s'|s,a)} [\Gamma_{n+1}^\pi V(s') - \Gamma_{n+1}^\pi V'(s')] \right] \right| \\
&\leq \max_s \mathop{\mathbb{E}}_{\pi(a|s)} \left[ \mathbb{1}_{a \neq a_n} |Q(s,a) - Q'(s,a)| + \mathbb{1}_{a = a_n} \gamma \mathop{\mathbb{E}}_{\mathcal{P}(s'|s,a)} |\Gamma_{n+1}^\pi V(s') - \Gamma_{n+1}^\pi V'(s')| \right] \\
&\leq \max_s \mathop{\mathbb{E}}_{\pi(a|s)} [\mathbb{1}_{a \neq a_n} \Delta + \mathbb{1}_{a = a_n} \gamma \Delta] \\
&\leq \max_s \mathop{\mathbb{E}}_{\pi(a|s)} [\mathbb{1}_{a \neq a_n} \Delta + \mathbb{1}_{a = a_n} \Delta] \\
&= \max_s \mathop{\mathbb{E}}_{\pi(a|s)} \Delta \\
&= \Delta.
\end{aligned}
$$

Finally,

$$
\begin{aligned}
\|\mathcal{T}^\pi Q - \mathcal{T}^\pi Q'\|_\infty &= \max_{s,a} |\mathcal{T}^\pi Q(s,a) - \mathcal{T}^\pi Q'(s,a)| \\
&= \max_{s,a} \left| \gamma \mathop{\mathbb{E}}_{\mathcal{P}(\cdot|s,a)} [\Gamma_1^\pi V(\cdot) - \Gamma_1^\pi V'(\cdot)] \right| \\
&\leq \gamma \max_{s,a} \mathop{\mathbb{E}}_{\mathcal{P}(\cdot|s,a)} |\Gamma_1^\pi V(\cdot) - \Gamma_1^\pi V'(\cdot)| \\
&\leq \gamma \max_{s,a} \mathop{\mathbb{E}}_{\mathcal{P}(\cdot|s,a)} \Delta \\
&= \gamma \Delta \\
&= \gamma \|Q - Q'\|_\infty.
\end{aligned}
$$

Since the discount factor $0 < \gamma < 1$, we have proved that $\mathcal{T}^\pi$ is a contraction mapping.

Importantly, this contraction holds for any historical trajectory $\tau$, even though this trajectory differs each time for an operator transform. Namely, every operator transform step will bring $Q$ and $Q'$ closer, *regardless of the actual value of the historical trajectory referred to*. Combining this with $Q^\pi$ being a fixed point of $\mathcal{T}^\pi$, we have shown that any $Q$ will converge to $Q^\pi$ if we repeatedly apply $\mathcal{T}^\pi$ to it.