# OpenReview forum: "TAAC: Temporally Abstract Actor-Critic for Continuous Control"
_NeurIPS.cc/2021/Conference — NeurIPS 2021 Poster_

### Official Review · Reviewer_Vncb · 2021-07-13

**Rating:** 7
**Confidence:** 4

**Summary:**

The paper proposes to add a component to the standard RL actor-critic architecture, that learns when to repeat (or not) the last action executed by the agent. This component is a simple binary classifier, conditioned on $s_t$, $a^-$ (the previous action) and $\hat{a}$ (what the actor wants to do now), that chooses between $a^-$ and $\hat{a}$. The authors theoretically motivate their algorithm, compare it to a wide range of related work, and provide extended experimental results that show that the proposed algorithm outperforms the Soft Actor-Critic, and its recent variations.

**Limitations And Societal Impact:**

This paper makes the effort of having a small paragraph about societal impact, which is nice, especially since there is not much to say given that this is a theoretical paper, that positively impacts the sample-efficiency of RL, but has no real-world problem in mind. I don't think that a larger discussion would be necessary.

**Main Review:**

The paper is well-written, and the idea it proposes is simple, in a good way. The experimental results are strong, and extremely well designed (many environments of many times, extra details in the appendix, well-motivated hyper-parameters). Overall, the contribution proposed in this paper has value, and, to me, clearly its place at NeurIPS.

Possible improvements to the paper could be as follows:

- In Section 4, two contributions are introduced at once: the introduction of the $\beta$ binary classifier, and the addition of $a^-$ as input to the actor. I may have missed it, but I did not see an ablation study that shows how the algorithm performs with only the $\beta$ classifier (leaving the actor as is), or only conditioning the actor on $a^-$ (without the binary classifier). I have the feeling that conditioning the actor on the previous action may help the agent with small amounts of partial observability (of full observability it has difficulties making sense of), by allowing the actor to choose an action at time-step $t$ that helps it make a decision at time $t+1$. I would therefore be very curious to empirically see the effect of that part of the contribution.
- The paragraph after Theorem 1 is a bit difficult to understand. I understand that it tries to explain how finding which "future action" is different from the current action is possible, even when the action is continuous, but I did not understand everything. To me, it seems that storing the actual choice of $\beta$ in the experience buffer, every time-step, allows a simple implementation of Equation 4 by just looking, in the buffer, at the first "future action t+n" for which $\beta(t+n-1) =$ choose new action. Is this what the paragraph tries to convey?

Author response: the authors give a motivation for why the actor always conditions on $a^-$, and say that they will do experiments without this conditioning, to see what impact it has. They will also make the paragraph after Theorem 1 clearer. Overall, this paper presents valuable work, and I recommend accepting it.

**Time Spent Reviewing:**

1

---

> ### Author Response · Authors · 2021-08-09
> **response to Reviewer Vncb**
>
> -   _**In Section 4, two contributions are introduced at once: the introduction of the**_ $\beta$ _**binary classifier, and the addition of**_ $a^-$ _**as input to the actor. I may have missed it, but I did not see an ablation study that shows how the algorithm performs with only the**_ $\beta$ _**classifier (leaving the actor as is), or only conditioning the actor on**_ $a^-$ _**(without the binary classifier). I have the feeling that conditioning the actor on the previous action may help the agent with small amounts of partial observability (of full observability it has difficulties making sense of), by allowing the actor to choose an action at time-step**_ $t$ _**that helps it make a decision at time**_ $t+1$_**. I would therefore be very curious to empirically see the effect of that part of the contribution.**_
>
> Conditioning on $a^-$ might indeed help in partial observations. However, we'd like to point out that except the **Driving** task, all other 13 tasks are fully observable. In these tasks, the current state can fully determine the optimal actions, without referring to previous history. Thus we don't claim addressing partial observability with $a^-$ as our contribution. (For the **Driving** task, we actually always added $a^-$ to $s$ in the experiments, for all the comparison methods. Even so, not all baselines perform well on that task.)
>
> In fact, the addition of $a^-$ as input to the actor was not intended to address partial observability. Instead, it's provided to the actor so that the actor could potentially output something $\hat{a}$ different from $a^-$, because there is a second stage that chooses between $\hat{a}$ and $a^-$. Although we haven't tried, we believe that keeping $\beta$ while removing $a^-$ from the actor inputs won't make much a difference, if we are not trying to address partial observability.
>
> -   _**The paragraph after Theorem 1 is a bit difficult to understand. I understand that it tries to explain how finding which "future action" is different from the current action is possible, even when the action is continuous, but I did not understand everything. To me, it seems that storing the actual choice of**_ $\beta$ _**in the experience buffer, every time-step, allows a simple implementation of Equation 4 by just looking, in the buffer, at the first "future action t+n" for which**_ $\beta(t+n-1)$ _**choose new action. Is this what the paragraph tries to convey?**_
>
> Mostly correct, except that at every step we need to look at both the stored repeating decision and the current switching decision, to see if either of them chooses new actions.
>
> It might be easier to understand that paragraph combined with Figure 2. Recall that we use $(a_n,b_n)$ to denote the stored actions in the replay buffer, and $(\tilde{a}_n,\tilde{b}_n)$ to denote the newly generated actions on sampled experiences. Starting from a sampled experience $(s_1,a_1,b_1)$ with its previous action as $a_0$, we compute the new actions with the current policy as $(\tilde{a}_1,\tilde{b}_1)$. If $b_1=\tilde{b}_1=0$, then we know both the behaviour policy and the learning policy choose to repeat at the same time step, so $a_1=\tilde{a}_1=a_0$. We can do the same thing for the next sampled experience $(s_2,a_2,b_2)$ on the same trajectory, and check if $b_2=\tilde{b}_2=0$. We stop at $n$ when either $b_n=1$ or $\tilde{b}_n=1$.

---

> > ### Comment · Reviewer_Vncb · 2021-08-25
> > **Re: response to Reviewer Vncb**
> >
> > Thank you for the clarifications.
> >
> > > we believe that keeping while removing from the actor inputs won't make much a difference, if we are not trying to address partial observability.
> >
> > I'm not complaining for this paper, but it happened to me very often that something I strongly believed ended up hiding something. So, now, even if I'm convinced that something does not make sense, or does not intervene on a result, I still do all the experiments I can think of (and often get surprised). In your case, removing $a^-$ from the input of the actor could still give surprising results, maybe even work much better than having $a^-$ as input (for reasons I don't see, but we never know).
> >
> > > [the explanation on the experience buffer]
> >
> > Thank you, this is much clearer now, I missed the fact that the current policy and $\beta$ are also used to see if it wants to change the action.
> >
> > Overall, I maintain my recommendation for acceptance.

---

### Official Review · Reviewer_VP6v · 2021-07-18

**Rating:** 6
**Confidence:** 3

**Summary:**

This work proposes a temporally abstract actor-critic (TAAC). They do so by incorporating a second-stage binary policy which chooses between the previous action and a new action - this act or repeat strategy hinges on the actually sampled action as opposed to the expected behavior. The key contributions are 1) a new compare-through Q operator for multi-step TD backup for policy evaluation and 2) computing the actor gradient by multiplying a scaling factor to the change in Q function wrt to the actions for policy improvement.


**Limitations And Societal Impact:**

Yes the authors have discussed societal impact. The limitations of the work are also covered in different sections of the paper - it would be valuable to consolidate them and add them in the conclusion section perhaps.

**Main Review:**

**Strengths**:
This work proposes a simple closed-loop off-policy RL algorithm TAAC, wherein a second stage chooses to act or repeat. The method shows how we can exploit the structure to maximize the PE-PI processes. The work proposed has the following strengths 1) The method is suitable for hybrid continuous-discrete action space including continuous control, 2) Enables persistent exploration and supports multi-step TD backups, and 3) The empirical results are strong in that the work is primarily empirical and shown to do well in a wide variety of tasks.

One key feature of the method which can be very useful is that it serves as a middle ground between one-level RL and HRL. The combination of ideas pitched allow this work to scale better to most baselines compared. The multi-step Q operator called compare-through is shown to facilitate efficient PE which is also shown to converge.


**Weaknesses**:

* A primary concern is how the proposed method deviates a lot from existing HRL methods which also consider action-repeats? Perhaps the key added value is the extension to continuous control for hybrid action space. It might be useful to elaborate on this in the discussion/related work. I would encourage you to expand the related work section to cover more HRL literature and discuss where the paper is positioned in the context of broader HRL methods.
* Arguably most HRL methods allow for the switching between the flat (one-step policies) and abstract actions (multi-step policies), especially the intra-option learning methods, which is the primary claim of this work. So what have we learned here which allows us to scale better to the list of tasks studied here?
* What happens when we don’t specifically want action-reproducible policies? Currently a lot of the improvements seem to stem from the nature of tasks where action-repeat might be beneficial.
* While the n-score and n-AUC is better as an average over tasks, TAAC is only marginally better or same as top baselines in most tasks as reported in the Figure 8: The unnormalized reward curves of the 14 tasks. It might be useful to expand on this in the main paper as to why you choose the evaluation metric to be n-score and n-AUC and consider averaging over all tasks as the primary metric.


**Empirical Analysis**:

The experiments evaluate TAAC in 5 categories of 14 continuous control tasks, covering simple control, locomotion, terrain walking (Brockman et al.,572016), manipulation (Plappert et al., 2018), and self-driving.

* The baselines are chosen fairly and compare a wide range of action-repeats methods. How do these methods compare to non-action repeat baselines? Any insights on this would be great. In particular, I am curious if the action that is being repeated is not encouraging exploration or performance, how does the method overcome such a choice?
* How many random seeds have been considered? Please mention it while reporting.


**Writing and Presentation**:
* The paper can be presented better in terms of explaining how this approach differs from existing approaches which consider action repeats. Consider a summary table which can highlight the novelty of the approach as compared to many other HRL off policy approaches as mentioned by the authors in Sec 1. Sec 2, and  Sec. 5.2.
* There are several insights in the more observations paragraph which is buried in 5.4. The paper might benefit from re-writing earlier parts to give some intuitions on key differences which this method brings to the table. For example 1)  Persistent exploration and the compare-through operator are crucial to the success of this approach. 2) The importance of the formulation of the closed-loop action repetition.


**Time Spent Reviewing:**

4

---

> ### Author Response · Authors · 2021-08-09
> **response to Reviewer VP6v (part 1)**
>
> -   **It might be useful to elaborate on this in the discussion/related work.** _**I would encourage you to expand the related work section to cover more HRL literature and discuss where the paper is positioned in the context of broader HRL methods.**_
>
> We agree that currently the difference between TAAC and other action-repeat methods might have been buried in the experiments section. We will extend the related work section to further elaborate on this.
>
> -   _**Arguably most HRL methods allow for the switching between the flat (one-step policies) and abstract actions (multi-step policies), especially the intra-option learning methods, which is the primary claim of this work. So what have we learned here which allows us to scale better to the list of tasks studied here?**_
>
> As noted in the introduction section,
>
> > _"Although this two-stage formulation seems straightforward, the key question we ask is: instead of training it with a generic actor-critic algorithm, how can we exploit its special structure to try to maximize the efficiency of policy evaluation and improvement, so that the overall algorithm yields strong sample efficiency and final performance? This will be the primary focus and contribution of the paper."_
>
> So our primary claim of this work is not the idea of repeating actions, instead it is: given that we are repeating actions, how can we instantiate this idea properly (among various possible ways), and how can we train it more efficiently? We agree that Line 38 might over-emphasize the action repetition idea itself. We will tone it down in the revision.
>
> As we've discussed earlier, and emphasized in our experiments, the reasons why TAAC scales better to the list of tasks are: (1) closed-loop repetition, (2) $\beta$ conditioning on newly generated $\hat{a}$, 3) compare-through operator, and 4) closed-form optimal $\beta^*$. (1) and (2) are about how we instantiate the action repetition idea, while (3) and (4) are the novel solutions to our instantiation. To our best knowledge, (1) has been studied only in (Neunert et al., 2020) before, and (2) has only been studied in (Biedenkapp et al., 2021) but for open-loop repetition. Of course, (3) and (4) are completely new solutions to action repetition.
>
> -   _**What happens when we don’t specifically want action-reproducible policies?**_
>
> We'd like to clarify that "action-reproducible" is not "action-repeatable". An action-reproducible policy could in theory produce the same action when visiting the same state twice. A discrete policy is naturally an action-reproducible policy because there is a limited number of discrete actions. A standard continuous policy is usually not. Our two-stage continuous policy is an exception for the reasons detailed in the paper.
>
> If we don't have an action-reproducible policy, the overall training method can still work, and we can just discard the compare-through operator. The performance will become a little worse (see the experiments on TAAC-1td vs TAAC-Ntd). However, even in this case, actions can still be repeated by $\beta$ for consecutive time steps during inference.
>
> -   _**Currently a lot of the improvements seem to stem from the nature of tasks where action-repeat might be beneficial.**_
>
> First, we'd like to clarify that we didn't pick these tasks by having action repetition in mind; instead TAAC manages to "mine" repeated actions from these tasks. In practice, as long as the action frequency cannot be pre-defined exactly to be the minimum that doesn't comprise the optimal control performance, there is always room for mining repeated actions.
>
> Besides this observation, please also see our detailed response to Reviewer SDFG on why we believe action persistence contributes to _good exploration_ in general (due to sparse reward or short-term deceptive dense reward). Even if a task's optimal control doesn't contain consecutively same actions,
>
> 1.   Action repetition in early stage encourages persistent exploration which makes the agent explore deeper and propagates rewards from future to present faster (Q value learned faster) with the compare-through operator (section 5.6.1 figure 3).
> 2.   When Q values are well learned, TAAC can dynamically adjust the repeating frequency depending on the action Q values (see different frequencies in Table 3). If a task indeed doesn't require action repetition for optimal control, then TAAC will frequently generate new actions instead of repeating.
>
> -   _**While the n-score and n-AUC is better as an average over tasks, TAAC is only marginally better or same as top baselines in most tasks as reported in the Figure 8: The unnormalized reward curves of the 14 tasks.**_
>
> An important observation should be made, that is, _**the top baselines in different tasks are different.**_ TAAC-1td is an ablation of our method, so excluding it, no method from SAC-Nrep, SAC-Krep, SAC-Hybrid can perform robustly across these tasks. While they might perform well in some tasks, they could fail very badly in other tasks. In order to address the robustness, we propose to look at the overall performance across tasks. And as reported, TAAC's overall n-score and n-AUC are *much* better than any baseline.
>
> Given a new task, unless one wants to go through the tedious process of trying various RL methods and picking the best one, we believe that our overall n-score and n-AUC are meaningful metrics to take into account.
>
> -   _**It might be useful to expand on this in the main paper as to why you choose the evaluation metric to be n-score and n-AUC and consider averaging over all tasks as the primary metric.**_
>
> Thanks for your suggestion. The reason why we consider n-score is that different tasks, even within the same group, can have vastly different reward scales (e.g., tens vs. thousands). So it is impractical to directly average their scores. It is not uncommon in prior works (Hessel et al., 2017; Cobbe et al., 2020) to set a performance range $[Z_0,Z_1]$ for each task separately and normalize the score of that task to roughly $[0,1]$ before averaging scores of a method over multiple tasks. Our n-score follows a similar idea. For n-AUC, as explained in the paper,
>
> > "we approximate n-AUC (area under the n-score curve normalized by x value range) by averaging n-scores on a n-score curve throughout the training. A higher n-AUC value indicates a faster convergence speed."
>
> It is a secondary metric to look at when two methods have a similar final n-score.
>
> By averaging n-score and n-AUC to get overall numbers, we want to evaluate the robustness of an RL method across many different tasks. It is not uncommon to see some RL methods perform well on a very narrow domain of tasks (perhaps specially designed), but very difficult to adapt to other domains. This situation is what we try very hard to avoid when designing TAAC and the evaluation metric.
>
> -   _**The baselines are chosen fairly and compare a wide range of action-repeats methods. How do these methods compare to non-action repeat baselines? Any insights on this would be great.**_
>
> Our baselines contain one non-action-repeat method which is vanilla SAC. From the experiment results, it performs worst regarding both n-score and n-AUC. For other general-purpose HRL methods that don't rely on action repetition, as noted in the paper:
>
> > While there exist many off-policy hierarchical RL methods that model temporal abstraction, ..., we did not find them readily scalable to our entire list of tasks (especially to high dimensional input space like CARLA), without considerable efforts of adaptation.
>
> -   _**In particular, I am curious if the action that is being repeated is not encouraging exploration or performance, how does the method overcome such a choice?**_
>
> As we pointed out in the response to Reviewer SDFG, we believe two important factors contribute to good exploration: diversity and persistence. Persistence is especially important in a sparse reward setting or given short-term deceptive dense rewards. Otherwise the agent will just wander around its initial position or discover sub-optimal solutions (Dabney et al., 2021). TAAC makes a good trade-off between diversity and persistence by dynamically adjusting repeating frequencies. So it is debatable that "action that is being repeated is not encouraging exploration", because repetition itself already encourages exploration from the perspective of persistence.
>
> On one hand, we impose an entropy target on $\beta$ so that it's encouraged to try both "repeat" and "not-repeat". On the other hand, even when a bad action is repeated many times, its Q value will quickly reflect the bad consequence (thanks to our compare-through operator for multi-step TD). Recall that our optimal form of $\beta$ will make decisions by comparing the Q values of repeating vs. no-repeating (Line 162 $\beta^*$). So $\beta$ will avoid repeating it again next time revisiting a similar state.
>
> -   _**How many random seeds have been considered? Please mention it while reporting.**_
>
> We use three random seeds throughout the experiments. We will move this information from the appendix to the main text.
>
> -   _**The paper can be presented better in terms of explaining how this approach differs from existing approaches which consider action repeats.**_
> -   _**The paper might benefit from re-writing earlier parts to give some intuitions on key differences which this method brings to the table.**_
> -   _**The limitations of the work are also covered in different sections of the paper - it would be valuable to consolidate them and add them in the conclusion section perhaps.**_
>
> We thank the reviewer for providing such detailed presentation suggestions. We will take them into account for the revision.

---

> > ### Comment · Reviewer_VP6v · 2021-08-20
> > **Thank you, revised score**
> >
> > Thank you for the detailed and rigorous response. Importantly, most of my confusions and concerns have been addressed. I must say I might have misunderstood parts of the submission in my previous attempt such as the terminology "action repeats" usage for instance.
> >
> > Having read the reviews of the other reviewers and rebuttal response, I have a much better understanding of the submission. In light of this, I have adjusted my score and confidence to reflect the changes in my rating.

---

> ### Author Response · Authors · 2021-08-09
> **response to Reviewer VP6v (part 2)**
>
> -   _**A primary concern is how the proposed method deviates a lot from existing HRL methods which also consider action-repeats?**_
>
> We believe that the paper has covered most HRL methods that also consider action repetition in our experiments. Table 1 provides a brief summary of the differences between TAAC and these methods.
>
> We'd like to point out that "consider action-repeats" is a very broad categorization. Under this category, there are various ways of how to formulate action repetition. For example, in the paper we present some baselines (SAC-Nrep, SAC-Krep, SAC-EZ) using _open-loop_ action repetition (output action and its duration together and commit to that duration). For closed-loop repetition, we compare with SAC-Hybrid whose repeat-or-act policy _doesn't depend on the newly sampled action_ $\hat{a}$_, and can only make the switching decision based on the expected behavior of_ $\pi_{\phi}$. Perhaps most importantly, we present new results for policy evaluation and improvement, both tailored to better learning closed-loop action repetition, and they greatly improve the final performance (as compared to TAAC-1td without the compare-through operator, and compared to SAC-Hybrid which learns an explicit $\beta$).

---

### Official Review · Reviewer_SDFG · 2021-07-18

**Rating:** 6
**Confidence:** 4

**Summary:**


This paper studies a single option: repetitive actions that can be selected via a two-layer architecture. First layer is a policy selected conditioned on previous action; second layer is a binary decision classifier to select the action by the 1st layer or the actual output at the current step. It is a very simple idea, with some seemingly flaws (see below), but perform well in a wide range of experiments.


**Main Review:**


Compare-through operator: This definition’s explanation is bit confusing. It looks this equation loops from the beginning time step to find all the subsequence of actions that match the target policy. However, line 125 doesn’t tell much except the operation notationwise. This looks the n will be usually small. Otherwise, your behaviour policy almost very close to the target policy, which becomes on-policy learning (easy).

Here you have a separate parameter for the target, apparently influenced by target network. Then I don’t get your Th. 1: what features does it deal with? If linear or tabular, why would you need the target “network parameter”? If it’s neural networks, then why Th. 1 is true? This is not a stable operator as far as 20 years of RL tells.

Exploration is an important aspect that this method may fail to address. Suppose one starts with a poor action. Then because of the binary decision on repeating it or not, there will be significant time repeating this single action in a long action sequence. Such method will only work with navigation in regular (rectangular) domains like Maze or Atari games. Regularizing with a joint entropy as in Section 4.3 may help, but I’m not convinced this is a good method at its heart for solving RL problems especially exploration is hard as we know.

Experiments compared with 6 baselines on 5 kinds of control tasks (all continuous action).
The “surprising” fact found in the experiment is the large portion of repeating actions in tasks. It is interesting to find TAAC is able be top performers; although one can argue that these tasks may be just suitable for repeated actions.

I think the empirical performance outweighs the seemingly inefficient exploration aspect, which is exactly the value  of this paper — a simple strategy with arguably inefficient exploration appearance but performing well in many tasks.



**Time Spent Reviewing:**

0.8

---

> ### Author Response · Authors · 2021-08-09
> **response to Reviewer SDFG**
>
> -   _**Compare-through operator: This definition’s explanation is bit confusing. It looks this equation loops from the beginning time step to find all the subsequence of actions that match the target policy. However, line 125 doesn’t tell much except the operation notationwise.**_
>
> "... to find all the subsequence of actions that match the target policy..." this should be the other way around. Given a sampled trajectory of length $N$ from the replay buffer, the compare-through operator takes an expectation, under the target policy at the sampled states, over all the subsequences (up to length $N$) of actions that match the sampled actions. Eq 4 is a point estimate of the full expectation. We will supplement line 125 with such an explanation.
>
> -   _**This looks the n will be usually small. Otherwise, your behaviour policy almost very close to the target policy, which becomes on-policy learning (easy).**_
>
> Indeed, $n$ is usually small. However, we would like to clarify that its value has no direct relation to on/off-policy training. Even with on-policy training where the behaviour policy is the same with the target policy, due to action sampling, there is no guarantee that they will output the same action at the same state. So on-policy training won't imply a large value of $n$. On the other hand, a large value of $n$could appear in off-policy training sometimes, because the fact that the behaviour policy is different from the target policy doesn't imply that they won't sample the same "repeat-or-not" decisions (possibly over the same trajectory of multiple steps) in a subset of state space. This can happen if both policies output very similar distributions on some states they are certain about.
>
> Note that the same issue of small $n$ might arise in other off-policy policy evaluation techniques like Retrace (Munos et al., 2016), although in a "soft" manner. There, the traces will be more likely to be cut as $n$ increases, when importance ratios decrease. See a comparison and relation between our compare-through operator and Retrace in Appendix K.1 (Line 751).
>
> We believe that the important finding in the paper is that even with a small or moderate value of $n$, our compare-through operator already helps a lot in the performance (see experiments on Ntd vs. 1td).
>
> -   _**Here you have a separate parameter for the target, apparently influenced by target network. Then I don’t get your Th. 1: what features does it deal with? If linear or tabular, why would you need the target “network parameter”? If it’s neural networks, then why Th. 1 is true? This is not a stable operator as far as 20 years of RL tells.**_
>
> This is a good question and we apologize for the confusion about Th1. Yes we can only prove the PE convergence in a tabular setting, as most deep RL algorithms do (e.g., SAC) for simplification. In a tabular setting, there is no longer "target network" (see Appendix K). We will revise Th1 to mention the tabular setting assumption.
>
> -   _**Exploration is an important aspect that this method may fail to address.**_
>
> We believe that there are two important aspects of good exploration:
>
> -   Diversity. The policy should produce a diverse set of actions to try out different possibilities.
> -   Persistence. After deciding an action, the policy should commit to it for a certain amount of time. The importance of persistent exploration has been emphasized by some prior works, for example Dabney et al., 2021, Grigsby et al., 2021.
>
> Diversity is usually thought of when talking about exploration. However, unless the reward is dense and short-term rewards are very informative about the optimal solution, only diversity won't lead to good exploration. When the reward is sparse and far from the initial state, diversity alone will make the agent wandering around its initial state because any persistent trajectory has an exponentially small probability. Even when the reward is dense, the short-term rewards could be deceptive and diversity alone will probably discover a suboptimal solution (Dabney et al., 2021). In contrast, persistence will make the policy explore deeper, which is useful in the above two scenarios. So we believe good exploration requires a trade-off between diversity and persistence, and it is debatable to say "this method may fail to address" exploration. Please see section 5.6.1 figure 3 for examples of how persistence improves exploration.
>
> We will include a short discussion about the importance of persistence in the introduction section.
>
> -   _**Suppose one starts with a poor action. Then because of the binary decision on repeating it or not, there will be significant time repeating this single action in a long action sequence.**_
>
> Repeating extremely bad actions for a long time could indeed hurt the diversity of the experience, since the replay buffer slots are always limited. There are actually several ways TAAC adopts to avoid this: First, the entropy of the repeating policy will make sure no extremely long action sequence will be generated in most if not all cases. Second, once the Q values are better learned, our derived optimal form of $\beta$will make decisions by comparing the Q values of repeating vs. no-repeating (Line 162 $\beta^*$). If the policy indeed repeats a bad action many times, its Q value will quickly reflect the bad consequence (thanks to our compare-through operator for multi-step TD). So $\beta$ will avoid repeating it again next time revisiting a similar state. Third, we can just set a hard limit to repeating times during rollout. Since TAAC is off-policy, this won't affect the training step.
>
> -   _**Such method will only work with navigation in regular (rectangular) domains like Maze or Atari games.**_
>
> As discussed earlier, because of reward sparsity and potential reward deceptiveness, persistence is also a good factor of good exploration. Our method could do well to trade diversity for persistence, so in general getting better exploration. Moreover, TAAC is proposed for _continuous_ control and there is no assumption about the shape of the domain or task type in our formulation. Our experiments on various continuous control tasks (including locomotion, manipulation, navigation, etc) also demonstrate that some seemingly unrelated tasks can also benefit from action repetition. This is potentially due to two reasons:
>
> 1.   Action repetition in early stage encourages persistent exploration which makes the agent explore deeper and propagates rewards from future to present faster (Q value learned faster) with the compare-through operator.
> 2.   When Q values are well learned, TAAC can dynamically adjust the repeating frequency depending on the action Q values (see different frequencies in Table 3). If a domain indeed doesn't require action repetition for optimal control, then TAAC will frequently generate new actions instead of repeating.
>
> -   _**Experiments compared with 6 baselines on 5 kinds of control tasks (all continuous action). The “surprising” fact found in the experiment is the large portion of repeating actions in tasks. It is interesting to find TAAC is able be top performers; although one can argue that these tasks may be just suitable for repeated actions.**_
>
> Indeed we found that some seemingly unrelated tasks surprisingly benefit from action repetition after training, for example, BipedalWaker and BipedalWalkerHardcore. As we think more about this observation, we realize that _in continuous control domain it is really difficult to find a task that is not "suitable" for repeated actions_. Unless the action frequency is pre-defined exactly to be the minimum that doesn't comprise the optimal control performance while leaving no room for temporal abstraction, action repetition is beneficial. It is worthwhile noting that we didn't pick these tasks by having action repetition in mind; instead TAAC manages to **"mine"** repeated actions from these tasks.
>
> In fact, even in our daily life, when we control our body to walk or manipulate, we tend to repeat actions in non-critical time steps. This is exactly what TAAC's trained policy discovers and our experiment results reveal.
>
> -   _**I think the empirical performance outweighs the seemingly inefficient exploration aspect, which is exactly the value of this paper — a simple strategy with arguably inefficient exploration appearance but performing well in many tasks.**_
>
> Again we would like to emphasize the role of persistence in good exploration. Only diversity alone is not enough. In this sense, our method doesn't have inefficient exploration; it just tries to strike a balance between diversity and persistence. This may help the reviewer explain why it performs well in many tasks.

---

### Official Review · Reviewer_M15z · 2021-07-19

**Rating:** 7
**Confidence:** 4

**Summary:**

The authors introduce TAAC, which stands for Temporally Abstract Actor Critic, an extension of the actor-critic framework, in the continuous control setting, to enable the policy to choose between selecting a new action or repeating the previous one. TAAC acts as a middle ground between hierarchical RL and “flat” RL, allowing the policy to perform temporal abstraction while not having to deal with the computational or complex formulation burden of hierarchical RL methods. TAAC requires only to change slightly the architecture of the actor-critic agent, i.e. replacing the policy had with a two-stage policy head and adding the previous action as input, and to change the policy evaluation and improvement update. In this setting, using off-policy correction for multi-step TD learning such as RETRACE is not applicable. Thus, the authors also introduce a compare-through operator to stabilize learning in this setting. The authors provide novel expressions of the policy evaluation and policy improvements updates, in this setting, as well as convergence proofs. Then the authors compare TAAC to 6 baselines and 1 ablation on a set of 14 benchmarks, grouped in 5 categories. They show that on average TAAC outperforms its competitors and especially shows a significant improvement on benchmarks that exhibit exploration difficulties and require planning over long horizons. Finally, the authors propose an in-depth study of the properties of their agents and notably show that it better cover the state space compared to SAC and also exploits often the action repetition leading to high performance even in settings where we would not expect it to be useful.

**Limitations And Societal Impact:**

This paper does not add limitations or potential negative societal impact to existing reinforcement learning methods.

**Main Review:**

I warmly thank the authors for their work.

Clarity:
- The paper is clear and well-written. I found it easy to read and to follow.

Soundness of the claims, significance and novelty of the contribution, and relevance to the NeurIPS community:
- The paper is well motivated and clearly positioned with respect to the literature
- The method section is very clear and the derivations are well presented and motivated
- The method seems fairly simple to implement and does not bring any additional computational cost, yet it seems to improve significantly the performance in many situations
- The experimental study is sound. The authors considered a wide range of benchmarks and thoroughly compared the method to baselines and ablations.
- The complementary analysis of the state space coverage and the frequency of re-used actions also make a lot of sense.

Limitations of this work:
- Line 88: the transition between the argument that computing $\rho^{\pi}(s)$ is usually intractable and the objective derived in the Equation is too fast and lacks maybe an argument. Also, $\mathcal{D}$ (which I assume is a replay buffer) is used in the Equation but not introduced.
- Line 186: The authors wrote: “to test if the algorithm generalizes”. I find this sentence misleading as when reading that I expect to see experiments about generalization in RL (generalization between environments, over initial states, etc …). I would recommend to change this sentence and maybe rather say something like: “to assess the performance of the algorithm”
- Line 248: I don’t get precisely what taking the argmax of the policy means. In the discrete action setting, I would understand it as choosing the action with higher probability, however, we are in the continuous action setting here. It would be great if the authors can clarify this point.

**Time Spent Reviewing:**

3

---

> ### Author Response · Authors · 2021-08-09
> **response to Reviewer M15z**
>
> -   _**Line 88: the transition between the argument that computing**_ $\rho^{\pi}(s)$ _**is usually intractable and the objective derived in the Equation is too fast and lacks maybe an argument. Also,**_ $\mathcal{D}$ _**(which I assume is a replay buffer) is used in the Equation but not introduced.**_
>
> Thanks for pointing this out. This is a Preliminaries section for background, and using the state distribution of $\mathcal{D}$ to approximate $\rho^{\pi}(s)$ is a common heuristic (as cited Lillicrap et al., 2016; Haarnoja et al., 2018). We will add one or two sentences for a clear explanation about this.
>
> -   _**Line 186: The authors wrote: “to test if the algorithm generalizes”. I find this sentence misleading as when reading that I expect to see experiments about generalization in RL (generalization between environments, over initial states, etc …). I would recommend to change this sentence and maybe rather say something like: “to assess the performance of the algorithm”**_
>
> We agree that the word "generalizes" might mislead readers into thinking of out-of-distribution generalization. What we really try to convey here is to see if TAAC is robust and can be readily applied to a wide range of tasks. We will rephrase this sentence as suggested.
>
> -   _**Line 248: I don’t get precisely what taking the argmax of the policy means. In the discrete action setting, I would understand it as choosing the action with higher probability, however, we are in the continuous action setting here. It would be great if the authors can clarify this point.**_
>
> We agree that the term "argmax" is not accurate enough. The more accurate description should be "taking the (approximate) mode of the policy".
>
> For a parameterized continuous policy, we are usually able to obtain or approximate its mode easily. In the paper, following SAC (Haarnoja et al., 2018) we use a _squashed_ diagonal Gaussian to represent the policy. Specifically, when sampling an action, we first sample from the unsquashed Gaussian $z\sim \mathcal{N}(\mu,\sigma^2)$, and then apply the squashing function $x=a\cdot tanh(z) + b$ to respect the action boundaries. However, because Gaussian is squashed, it's difficult to exactly obtain its mode due to action boundary effect, and the mode is no longer $\mu$ near the boundaries (it will be pushed towards the center of the action interval). So in practice, to approximately get the mode, we first get the mode $\mu$ from the base Gaussian, and then directly apply the squashing function $\tilde{\mu}=a\cdot tanh(\mu) + b$. This $\tilde{\mu}$ is treated as the mode of the policy.
>
> We will update the "argmax" related descriptions in the main paper and add a paragraph to further explain this in the Appendix.

---

### Author Response · Authors · 2021-08-09
**Summary of response**

We thank reviewers for their time and constructive comments.

Among many replies, we mainly emphasize that besides action diversity, _persistence_ is also a very important contributor to efficient exploration (Dabney et al., 2021; Grigsby et al., 2021). Although repeated actions might decrease diversity to some extent, TAAC (dynamically) trades diversity for persistence and obtains good exploration overall. Thus it is debatable to say that our method is inefficient in exploration. In fact, to improve exploration efficiency is one of our main motivations of modeling action repetition in TAAC (section 5.6.1 figure 3).

Another major reply is to address whether our tasks are just suitable for action repetition. Note that even some seemingly unrelated tasks greatly benefit from action repetition, e.g., BipedalWalker, BipedalWalkerHardcore, LunarLanderContinuous, etc. We didn't pick these tasks by having action repetition in mind; instead TAAC manages to **"mine"** repeated actions from these tasks. That's exactly an important finding revealed by this paper: unless the action frequency is pre-defined exactly to be the minimum that doesn't comprise the optimal control performance while leaving no room for temporal abstraction, action repetition is beneficial. Even a task doesn't require action repetition at all, persistent exploration in the early stage of training will also benefit the training and improve performance (section 5.6.1).

Please see our detailed responses to each reviewer. We hope that they have addressed the reviewers' questions and concerns. Please let us know any further questions.

**References in the responses:**

Dabney et al., 2021, Temporally-extended epsilon-greedy exploration, ICLR

Grigsby et al., 2021, Towards Automatic Actor-Critic Solutions to Continuous Control, arXiv

Lillicrap et al., 2016, Continuous control with deep reinforcement learning, ICLR

Haarnoja et al., 2018, Soft actor-critic algorithms and applications, arXiv

Munos et al., 2016, Safe and efficient off-policy reinforcement learning, NeurIPS

Neunert et al., 2020, Continuous-discrete reinforcement learning for hybrid control in robotics, CoRL

Biedenkapp et al., 2021, TempoRL: Learning When to Act, ICML

Hessel et al., 2017, Rainbow: Combining Improvements in Deep Reinforcement Learning, AAAI

Cobbe et al., 2020, Leveraging Procedural Generation to Benchmark Reinforcement Learning, ICML

---

### Decision · Program_Chairs · 2021-09-27

**Decision:**

Accept (Poster)

**Comment:**

The discussion helped clarifying points that were misunderstood by some reviewers initially. It also helped improving the clarity of the paper. There is a consensus on the scores and, now, the contribution is regarded as significant and simple. The method performs very well, it is simple enough to be implemented easily and it is computationally neutral.